# Acquisition of non-olfactory encoding improves odour discrimination in olfactory cortex

Noel Federman [1,3] ✉, Sebastián A. Romano [1,3] ✉, Macarena Amigo-Duran [1,2], Lucca Salomon [1,2] & Antonia Marin-Burgin [1] ✉

Olfaction is influenced by contextual factors, past experiences, and the animal's internal state. Whether this information is integrated at the initial stages of cortical odour processing is not known, nor how these signals may influence odour encoding. Here we revealed multiple and diverse non-olfactory responses in the primary olfactory (piriform) cortex (PCx), which dynamically enhance PCx odour discrimination according to behavioural demands. We performed recordings of PCx neurons from mice trained in a virtual reality task to associate odours with visual contexts to obtain a reward. We found that learning shifts PCx activity from encoding solely odours to a regime in which positional, contextual, and associative responses emerge on odour-responsive neurons that become mixed-selective. The modulation of PCx activity by these non-olfactory signals was dynamic, improving odour decoding during task engagement and in rewarded contexts. This improvement relied on the acquired mixed-selectivity, demonstrating how integrating extra-sensory inputs in sensory cortices can enhance sensory processing while encoding the behavioural relevance of stimuli.

Olfaction stands out among sensory modalities as a complex perceptual process that is heavily influenced by learned associations, contextual factors and the animal's state[1–3]. These diverse sources of information could be integrated into the olfactory pathway to successfully guide olfactory-cued decisions. The piriform cortex (PCx), the largest region of the mouse primary olfactory cortex, plays a central role in collecting odour information from the olfactory bulb to encode odour identity[4–6]. Nevertheless, odour representations in the PCx drift over the course of days[7], suggesting that the PCx is a fast and continual learning system. Additionally, the PCx receives extensive inputs from higher-order processing areas[8–12], which may contribute other types of information to PCx neuronal representations. Indeed, given its anatomical organisation and the broad extension of its plastic recurrent connections, the PCx has been proposed to parallel the integrative function observed in associative cortices[8,10,13,14].

Integrating sensory, behavioural and contextual information at the initial stages of cortical odour processing offers potential advantages. Rather than simply relaying segregated information to higher-order association brain areas for subsequent processing, early integration at the primary olfactory cortex could enable state-dependent olfactory representations, allowing animals to flexibly process information according to their current behavioural demands[15]. A characteristic signature of such integration at the single-neuron level is the presence of mixed-selectivity neurons that simultaneously encode multiple task variables of diverse nature[16,17]. While mixed-selectivity is well-documented in higher-order regions like the prefrontal cortex[16,18,19] and hippocampus[20], its impact on sensory computations in

[1]Instituto de Investigación en Biomedicina de Buenos Aires (IBioBA)-CONICET-Partner Institute of the Max Planck Society, Godoy Cruz 2390, C1425FQD Buenos Aires, Argentina. [2]Universidad de Buenos Aires, Facultad de Ciencias Exactas y Naturales, PhD Program, Buenos Aires, Argentina. [3]These authors contributed equally: Noel Federman, Sebastián A. Romano. ✉e-mail: nfederman@ibioba-mpsp-conicet.gov.ar; sromano@ibioba-mpsp-conicet.gov.ar; aburgin@ibioba-mpsp-conicet.gov.ar

primary sensory cortices has received limited research attention. Non-sensory modulations of neuronal activity in primary sensory cortices have been found[21–28], including the observation of spatial maps in PCx[28]. However, how they affect sensory function is not yet understood.

Here we study how encoding in PCx is affected when odours acquire behavioural relevance through associative learning between odorants, spatial contexts and rewards. We found that learning leads to the encoding of non-olfactory information in the mouse PCx through reshaping its functional organisation into a multidimensional and mixed-selective scheme. Moreover, we observed that the acquisition of mixed-selectivity improves decoding of odour identity from the PCx neuronal ensembles.

## Results

### Mice learn to discriminate the same odour in different spatial contexts

We developed a behavioural task in which mice had to learn that a particular odour is rewarded when presented at the entry of a specific visual context (Fig. 1a and Supplementary Fig. 1a, b). Thirsty animals were trained to traverse a virtual linear corridor in order to reach a spatial location where one of two distinct visual contexts could be presented (green or grey). When entering the context, they received a 1-second-long puff of one of two possible odours (isoamyl acetate or ethyl butyrate). When reaching the reward zone at context exit, mice could choose to lick (GO response) or not to lick (NO-GO response) a reward spout to trigger the delivery of a water drop. Only one of the four possible odour-context combinations would lead to water delivery (rewarded odour-context; $O_R C_R$; Fig. 1b). Mice had to learn to discriminate the target $O_R C_R$ association with GO responses only on those trials. Importantly, the behavioural task was designed so that while mice ran through the virtual corridor, they were able to see the approaching visual context in anticipation of odour delivery (Fig. 1a and Supplementary Fig. 1b). This allowed us to evaluate how contextual and olfactory information are individually related to behaviour and neuronal activity. Animals became experts after around 7 days of training (Fig. 1c). By analysing several task-related behavioural variables, such as licking responses, locomotion speed and inhalation rate,

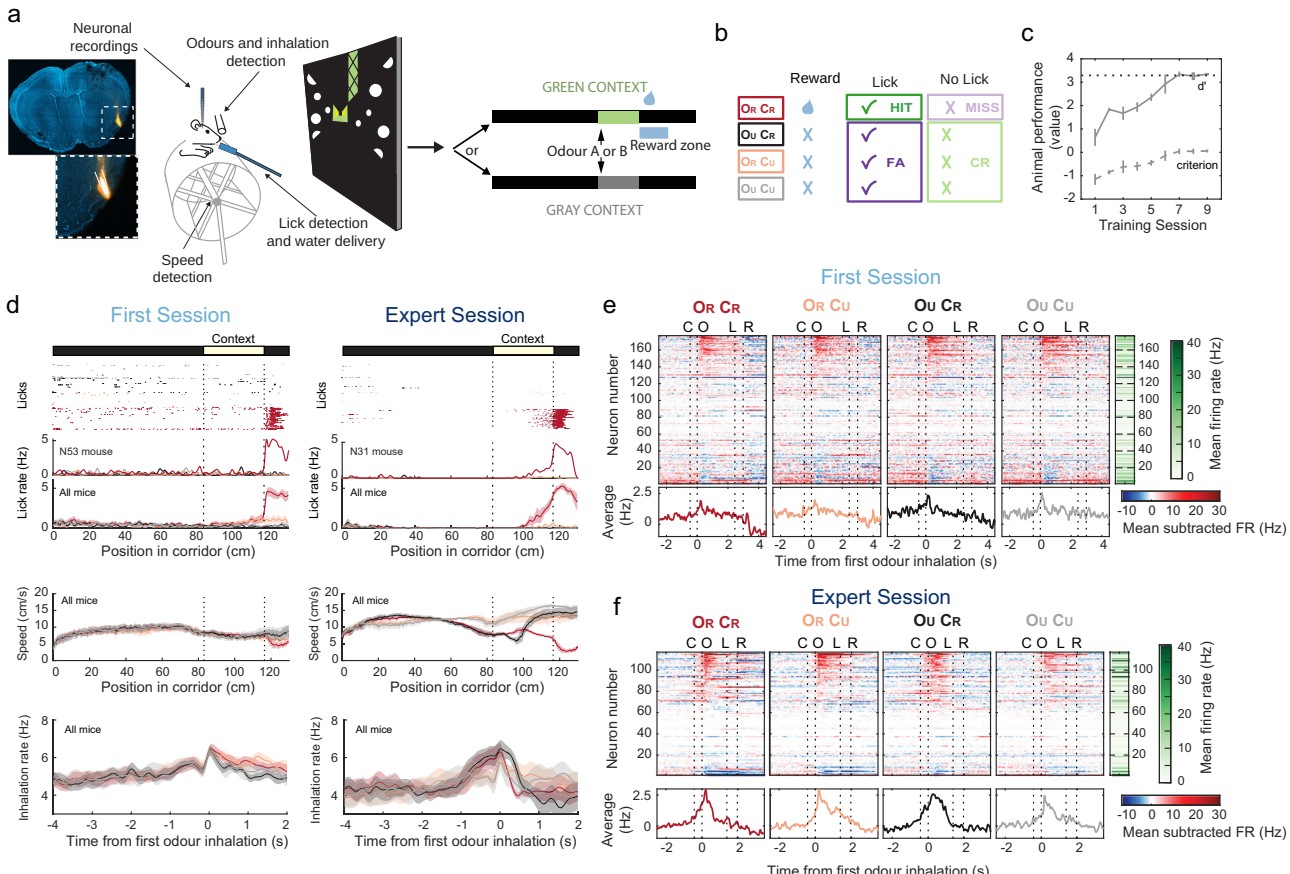

**Fig. 1 | Learning an associative odour-virtual context-reward task. a** Schematic of the virtual reality setup. Example of a recording site with the position of the silicon probe stained with dye I. Mice can run through a corridor with two alternative visual contexts and odours. Delivery of odours occurs after entering visual context and the reward delivery occurs at the exit. **b** Only one combination of context and odour is associated with the water reward. $O_R$ and $O_U$, Odour rewarded and unrewarded. $C_R$ and $C_U$, visual context rewarded and unrewarded. Depending on context, red/pink or black/grey represent different odours. CR correct rejection, FA false alarm. **c** Behavioural performance (mean values ± SEM of behavioural $d'$ and criterion, see 'Materials and methods'). Dashed line corresponds to 95% hits with 5% FA responses ($n = 10$ for first session and $n = 4$ mice for the rest of the training sessions). **d** Changes in behavioural variables through learning. Top rasters show licks (dots) aligned to position in the corridor of two example animals (first session and expert session) where trial type is colour coded as in (**b**). Lower panels show average (±SEM) licking rate, speed and inhalation rate on different trial types for first session ($n = 6$ mice) and expert animals ($n = 4$ mice). Colour maps showing all neuronal responses as variation around the neuron's mean FR (we subtracted the mean FR from the actual FR) in first session (**e**) and expert session (**f**) animals aligned to the time of the first odour inhalation. Vertical dashed lines indicate alignments to median times of the following events: C context entry, O odour delivery, L licking response before water delivery, R reward delivery. Mean-subtracted FR stitched for visualisation (see 'Materials and methods'). Green colour-coded column on the right shows the mean FR of each neuron. Neurons were sorted according to their average responses to $O_R C_R$ trials. Lower panels show the average mean subtracted FR across the neuronal population for each trial type. Source data are provided as a Source Data file.

we found that animals adjusted their behaviour following learning. Animals in the first session of training had similar licking responses, running speed and inhalation rate in the four types of trials before receiving the reward, in which case licking rate increases in $O_R C_R$ trials at the reward zone (Fig. 1d). On the other hand, learning induced anticipatory behaviours in expert animals, such as an anticipatory increase in licking responses when approaching the reward zone only in $O_R C_R$ trials and an increase in sniffing rate before odour delivery only in rewarded contexts ($C_R$ trials) (Fig. 1d), indicating that animals have learned the association between odour-context-reward. In addition, running behaviour changes after learning in the different types of trials: animals slowed down when approaching a possible rewarded context ($C_R$ trials), then increased speed if the odour is unrewarded ($O_U C_R$) or stopped to lick for reward in the rewarded condition ($O_R C_R$). These changes in several task related variables indicate that expert animals use odour in combination with visual contexts but also corridor position to guide their behaviour.

## Population dynamics in piriform cortex reveals spatial and associational signals that emerge following learning

To investigate whether PCx neural responses change as animals learn to associate odours with visual contexts and rewards, we performed acute recordings of PCx neuronal activity from animals in their first training session (177 neurons from 6 animals, $29.5 \pm 21.1$ neurons per animal; mean $\pm$ s.d) and animals that reached expert performance (117 neurons from 4 animals, $29.2 \pm 23.6$ neurons per animal; mean $\pm$ s.d.; Fig. 1a, e, f). There were no significant differences in the firing rates recorded in both conditions (Supplementary Fig. 1c). PCx neurons showed typical excitatory and inhibitory odour responses locked to odour inhalation onset (Fig. 1e, f).

To analyse if we could extract odour information from these recordings at the population level, we cross-correlated the activity between trial types in first-session and expert animals. We pooled recordings and calculated a time series of population activity vectors containing the spike counts of each cell, computed in a sequence of time bins around odour stimulation. The similarity between these vectors evoked by different trial types and at different times during first sessions and expert sessions was quantified by the Pearson correlation coefficient. First-session evoked activity patterns following odorant stimulation that distinguished odour identity (Fig. 2a, two diagonal blocks in top panel). Analysis of activity patterns from expert animals (Fig. 2a, bottom panel) points to a more complex scenario: trial types can be initially grouped by visual context as animals approach the context entry (Fig. 2a, checkerboard pattern emerging before odorant stimulation in expert animals), later response patterns discriminate between odours (Fig. 2a, diagonal blocks around 0.3 s in expert animals), and by 1.2 s after first odour inhalation they fully differentiate rewarded trials from the rest of conditions (Fig. 2a, $O_R C_R$ trials in expert animals). We obtained a comparable result through principal component analysis (PCA). Projecting the time series of evoked activity patterns from expert recordings in principal component space resulted in trajectories that diverge first according to contexts, then to odours, finally leading to a deviation of rewarded trials trajectory (Fig. 2b). These results indicate that non-olfactory signals, presumably related to visual context and reward, can be observed in the activity of the population of PCx neurons. Interestingly, those responses were acquired with learning. Despite the fact that both first-session and expert animals are able to see the approaching visual contextual cues from a distance, the activity of neurons from first-session animals does not group by visual contexts before context entry, suggesting that learning induces a contextual modulation of PCx activity before the potentially rewarded odour arrives. This contextual modulation in expert mice does not seem to result from licking (anticipatory licking is observed in $O_R C_R$ trials exclusively, and only after odour onset) or changes in running speed

between $C_R$ and $C_U$ trials (which are moderate before context entry; Fig. 1d), and are probably related to a heightened arousal triggered by $C_R$.

Accordingly, by calculating the number of principal components needed to explain total signal variance in odour-aligned PCx population activity, we found that first-session odour responses could be explained by 2 dimensions, while expert-session responses needed 5 dimensions (0.011 and 0.043 dimensions per recorded neuron, respectively, Supplementary Fig. 2g), further showing that the space of possible response patterns has higher dimensions in expert recordings.

To quantify which of the task-related variables significantly contribute to PCx neuronal activity, we decomposed the time-varying contributions of sensory stimuli conditions to the population activity dynamics by applying demixed PCA (dPCA[29]) across odour-context combinations (Fig. 2c and Supplementary Fig. 2a, c; 'Methods'). dPCA takes into consideration the different task conditions to infer components that depend on single parameters, allowing to disentangle data dependencies on specific variables, like odours, contexts and reward. In both first session and expert animals, signal variance had strong contributions from odour components (Fig. 2c, pie charts) and odour information can be decoded from several components that show either transient or sustained activity (Fig. 2c and Supplementary Fig. 2b, d). Contributions from contextual and context-odour interaction components were larger in expert animals (Fig. 2c, pie charts), and contextual information could be decoded from components that anticipate the entry and exit of the visual context zone (Fig. 2c and Supplementary Fig. 2b, d). Again, this is due to the fact that animals can see the approaching visual contextual cues from a distance and this is reflected in the separation of some of the component curves in anticipation of $C_R$ and $C_U$ entrance (Fig. 2c component #9, Supplementary Fig. 2d component #3, #9). Reward information is observed as a peak of contextual and context-odour interaction components in $O_R C_R$ trials after reward consumption in both first-session and expert animals. Nevertheless, in expert animals, some components could discriminate $O_R C_R$ trials before the animal's decision to lick for reward (Fig. 2c and Supplementary Fig. 2d, see components #3, 4 and 12 in Experts). Interestingly, dPCA of Hit and False alarm trials show similar trajectories before the decision to lick for reward, which then deviate during reward consumption (Supplementary Fig. 2e). Altogether, these observations indicate that, only after learning, population PCx activity encodes information about spatial and associational aspects of the task. How are these population representations composed?

## Learning induces mixed-selectivity encoding in single neurons of piriform cortex

The observed olfactory and contextual signals could be conveyed by different neuronal sub-populations of PCx neurons. Alternatively, this information could be merged by single neurons. We evaluated this by studying single PCx neuronal responses from first-session and expert animals performing the task (Fig. 3a). As expected by previous work[4,30,31], PCx neurons showed either excitatory or inhibitory responses that depended on odorant identity, and displayed respiratory-locked activity (Figs. 1e, f and 3b, c and Supplementary Fig. 3d, h). We also found neurons whose activity levels varied according to the animal's running speed, along with changes in firing rate preceding and/or following single licking events across all trial types and following reward consumption in $O_R C_R$ trials (Fig. 3b, c and Supplementary Fig. 3b, c, f, g). Interestingly, once animals learned the task, we found neurons that fired according to the animal spatial location along the virtual corridor (Fig. 3c, 'Position responses') and neurons that displayed a diversity of discriminative responses to the different odour-context combinations (Fig. 3c and Supplementary Fig. 3i, 'Associative responses'). Some of these neurons were selective for the rewarded $O_R C_R$ trials following odorant stimulation but in

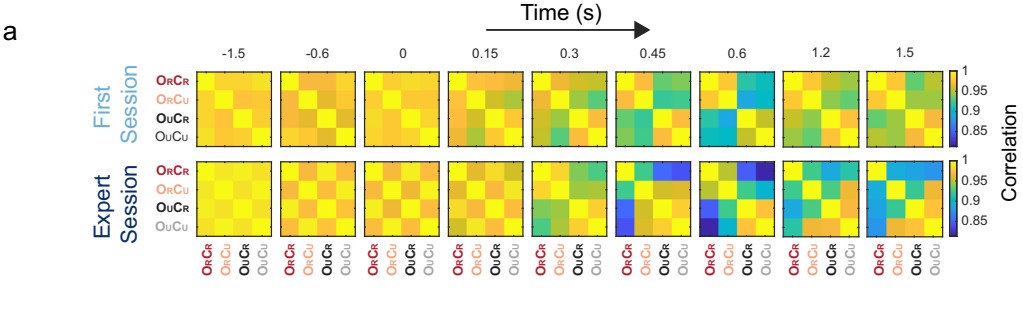

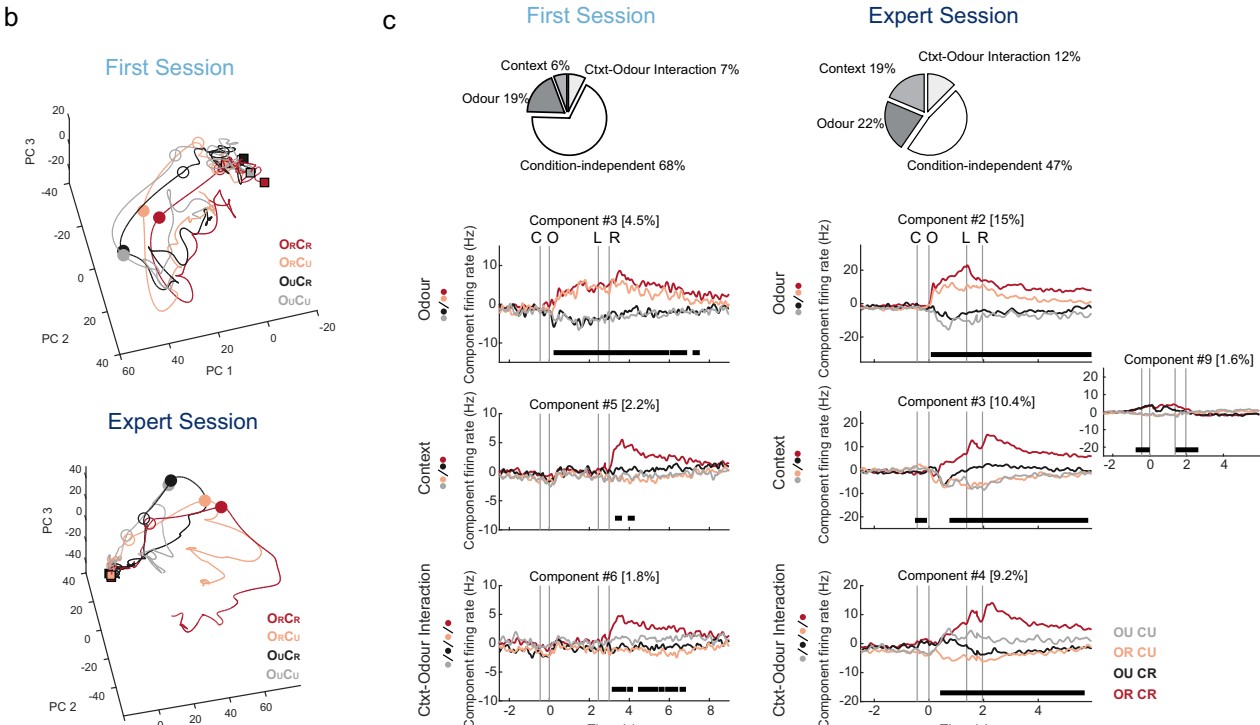

**Fig. 2 | Population dynamics in the piriform cortex reveals spatial and associational signals. a** Pearson correlation coefficients of the time series of population activity vectors containing the spike counts of each cell, computed in a sequence of time bins around odour stimulation, for each trial type. **b** Principal component analysis (PCA) of the pooled time series of evoked activity patterns for first-session and expert-session recordings. Filled squares, empty circles and filled circles label −2.5 s, 0 s and 0.2 s relative to first odour inhalation, respectively. Colours indicate trial type. **c** Demixed principal components (dPCA). Pie charts show how the total signal variance is split among task parameters. Top panels: first odour discrimination components, middle panels: first context discrimination

components (inset in expert session also shows the second component), bottom panels: first context-odour interaction component. First-session recordings on the left, and expert-session recordings on the right. In each subplot, the full data are projected onto the respective dPCA decoder axis. Thick black lines show time intervals during which the respective task parameters (odour, context, and context-odour interaction) can be significantly decoded from single-trial activity (see 'Materials and methods'). Variances explained by each component are shown as percentages. More dPCA components are shown in Supplementary Fig. 2. Source data are provided as a Source Data file.

anticipation of the animal's licking response (Fig. 3c and Supplementary Fig. 3i). In the time window preceding this decision to lick there is presumably a combination of pre-motor and reward anticipation signals which are hard to discriminate. Nevertheless, we were able to find neuronal responses that specifically preceded licks performed after odour stimulation and in anticipation to the reward zone during GO trials, but not to licks performed before odour delivery (i.e., 'spontaneous' licks unrelated to any reward-predicting odour-context association), indicating a clear distinction between reward anticipation and motor preparation neuronal responses (Supplementary Fig. 4). These results show that beyond odour identity encoding, the activity of individual neurons in PCx is rich in behavioural information, capable of context-dependent olfactory processing and displays learning-related associative and choice signals.

To characterise the mixture of sensory and behavioural signals that simultaneously affect the activity of single neurons, we

implemented a statistical approach based on a Poisson generalised linear model (GLM) of neuronal encoding[32,33] (Fig. 4a and Supplementary Fig. 5a, b; 'Methods'). The model fits single-trial time-varying spike responses against the timing and value of each task variable during each trial (Supplementary Fig. 5a). For each neuron, we selected an encoding model containing the combination of these variables that maximised model performance in predicting neuronal activity from trials held out from the fitting procedure (Supplementary Fig. 5c). In doing so, it infers linear filters (or kernels) that quantify the dependencies of neuronal responses on each variable (Fig. 4b–d and Supplementary Figs. 6a, b and 8). The variables considered in our model were sensory, motor and cognitive: rewarded and unrewarded odours ($O_R$ and $O_U$); animal position along rewarded and unrewarded visual contexts ($C_R$ and $C_U$); reward consumption (R); inhalation (I); running speed (S); licking (L); modulation of odour responses by presence of rewarded context ($modO_R$ and $modO_U$); anticipation to GO response

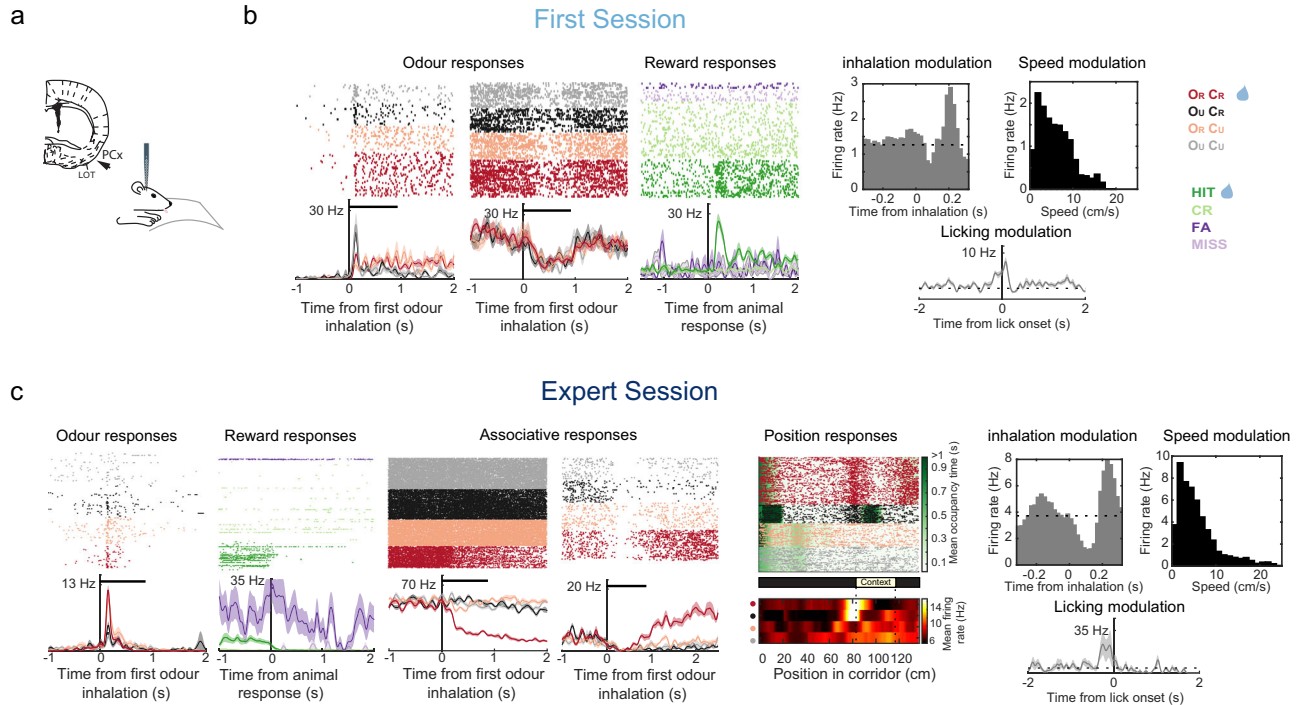

**Fig. 3 | Single-cell responses across learning. a** Schematic of PCx recording site in behaving animals. **b** Examples of neurons from mice recorded during first sessions. Odour and reward responses are shown in raster plots of action potentials (ticks) colour coded by trial type (odour responses) or trial outcome (reward responses), according to the colour code on the right. Odour pulse duration is indicated with a horizontal black bar. Bottom panels show the average firing rate of the neurons (mean values ± SEM). Modulation by inhalation, speed and licking are also shown in the right three panels. **c** Examples of neurons from mice recorded during expert sessions. Odour, reward, inhalation, speed and licking responses are shown as in (**b**). Expert animals also show associative responsive neurons as well as neurons responding to the animal position along the corridor. Position responses in the bottom panel are normalised by the time the animal spent in each location (animal's mean occupancy time is shown in green under the raster plot).

(preGO). We also included a bias kernel to account for differences in the baseline firing rate of neurons. For each neuron, we selected an encoding model containing the combination of these variables that maximised model performance. Using simulations, we confirmed the validity and accuracy of the GLM model for capturing with precision the individual contributions to neuronal responses of stimulus, behavioural and task-related variables (Supplementary Fig. 7; see 'Methods'). Model predictions of peri-stimulus time histograms (PSTHs) aligned to task events (Supplementary Fig. 8) explained a larger degree of variance of pooled expert-session data (66%) compared to pooled first-session data (24%), even after matching the total number of trials between both conditions (Experts: 49%, First-session: 23%). This is in line with the previous observation that individual stimulus and contextual dPCA components better capture variance in population data from expert-session recordings than from first session (Supplementary Fig. 2a, c). Together, these results point to a stronger and more reliable modulation of PCx neuronal activity by sensory and behavioural variables after learning. The lower reliability of responses observed in first session animals could also be related to a less stereotyped behaviour and underlying neuronal processes due to mice being exposed for the first time to the task.

The fitted models reveal that single PCx neurons carry information about a variety of olfactory and non-olfactory variables in distinct ways, differing in magnitude and temporal dynamics (Fig. 4b and Supplementary Figs. 6 and 8). The neuronal activity of first-session animals was dominated by inhalation and odour kernels (Fig. 4c). At the level of the fitted kernels, learning is accompanied by two prominent features: the presence of strong modO$_R$ and modO$_U$ kernels that modulate odour responses in a context dependent manner a few hundred milliseconds following olfactory stimulation, and the incorporation of C$_R$ and C$_U$ kernels that encode the entry and exit cues of the visual contexts (Fig. 4b, c). Neurons from expert animals were modulated by a larger number of parameters. This increase in multiplexing after learning was confirmed by estimating the percentage of neurons modulated by each variable (Fig. 4d), the absolute and relative contribution of the variables to the model (Fig. 4c, right panel and Supplementary Fig. 5d), and the amplitude of the obtained kernels (Fig. 4b and Supplementary Fig. 5e). For all these quantities, expert-session neurons showed a larger extent of visual context encoding, modulations of odour responses by rewarded context, and activity in anticipation to GO responses. In addition, licking and reward consumption had larger absolute contributions in experts, while reward consumption also showed larger kernel amplitude (Supplementary Fig. 5d, e). Thus, learning embeds piriform cortex neurons with mixed-selectivity responses to different aspects of the behavioural task.

## Learned mixed-selectivity responses on PCx are structured

We investigated whether learning reorganises PCx neurons in new functional groups. The mixed-selective neurons observed after learning could be selective to any given mixture of variables, with no particular preference. On the contrary, learning could promote specific patterns of mixed selectivity composed of particular combinations of variables that modulate together single-neuron responses. To address this, we used a hierarchical clustering approach to identify assemblies of neurons with similar modulation profiles ('Methods'). Clustering neurons according to the relative contributions of the variables they encode (Fig. 4e and Supplementary Fig. 9) showed that first-session profiles were typically dominated by contributions from single variables (Fig. 4e, bottom row, 47.8% of fitted cells), or pairs of variables that included either inhalation or odour kernels (Fig. 4e, upper rows of 'Inhalation' and 'Odour' columns, 24.3% of fitted cells). Clusters of expert-session neurons were less dominated by inhalatory drive and

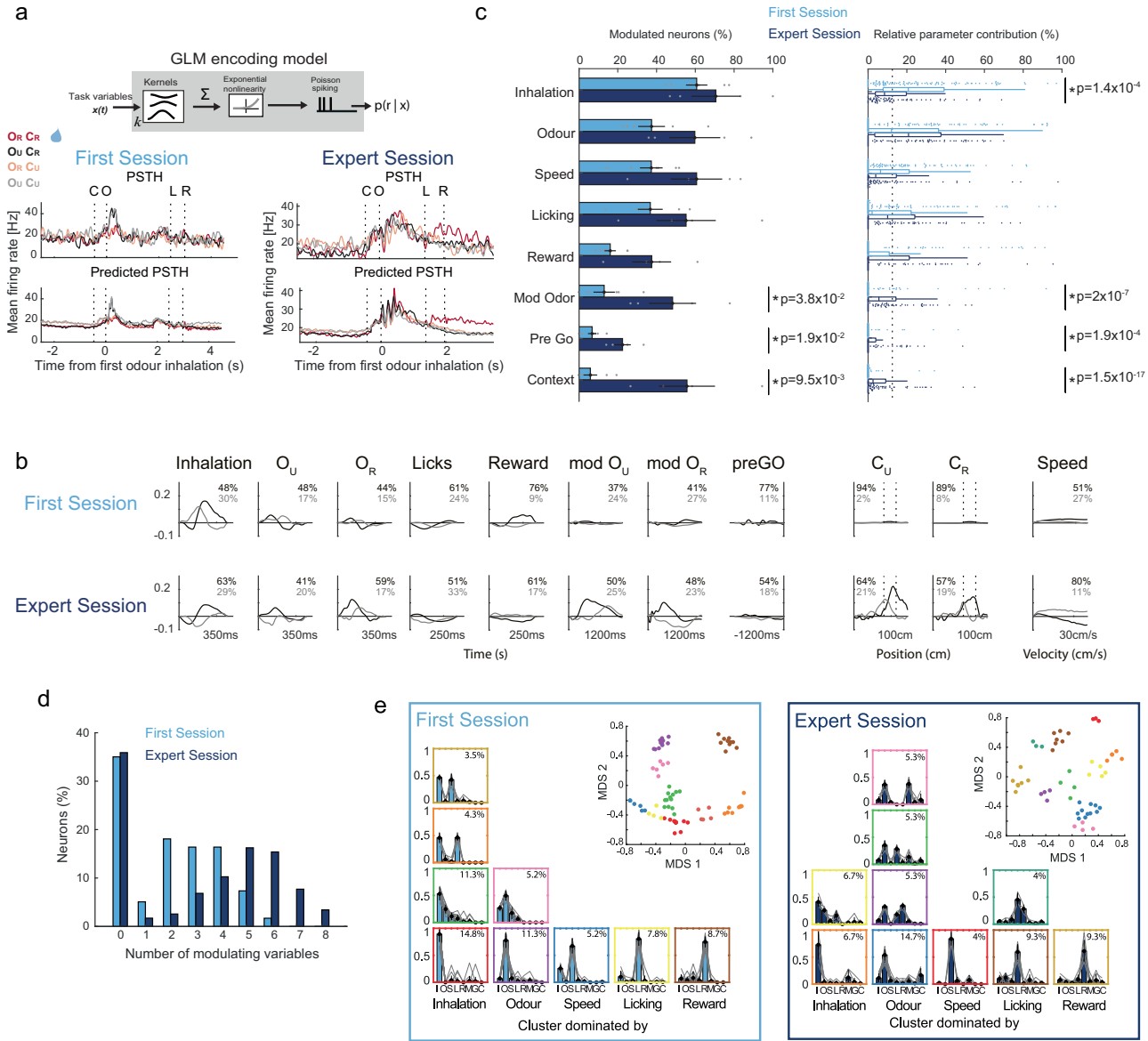

**Fig. 4 | Learning induces multidimensional encoding in piriform neurons.**
**a** Top, GLM applied to single-neuron single-trial recordings. Bottom, examples of peri-stimulus time histograms (PSTH) and model-predictions of PSTHs of one first-session (top) and one expert-session (bottom) neuron. **b** First (black) and second (grey) PCA component of pooled kernels for inhalation, unrewarded and rewarded odour ($O_U$ and $O_R$), licking, reward, modulation of context onto $O_U$ or $O_R$ (modO$_U$ and modO$_R$), activity before a GO decision (preGO), unrewarded and rewarded context ($C_U$ and $C_R$) and animal speed. Explained variance by each component is indicated in percentages, first-session recordings (top) and expert-session recordings (bottom). **c** Left, percentage of modulated neurons per animal for each task variable. Bars indicate mean values ± SEM, grey dots are single animals ($n = 6$ and $n = 4$ for first-session and expert animals). Right, relative contribution of task variable to neuronal modulation. Dots are single neurons ($n = 177$ and $n = 117$ for first-session and expert animals), summarised in box plots (boxes: medians and interquartile ranges, whiskers to the most extreme non-outlier datapoints). Dashed line marks equal contribution by all variables. Asterisks indicate statistically

significant differences between first (light blue) and expert (dark blue) session recordings (two-sided Wilcoxon rank sum test). **d** The number of variables modulating PCx neuronal activity increased with learning (two-sided Wilcoxon rank sum test for tied data; $p = 1.1 \times 10^{-4}$). **e** Clustering GLM-fitted neurons ($n = 114$ and $n = 74$ for first-session and expert animals) according to the relative contribution of their encoded task variables. Bar plots in coloured squares represent the mean relative contribution of the variables to each individual cluster (mean ± std). Grey lines show individual neurons in the cluster. Percentages indicate the proportion of neurons included relative to the total population. Columns of bar plots are sorted according to the dominating variable (labelled below each column, I inhalation, O odour, S speed, L licking, R reward, M modulation of odour by context, G Pre Go modulation, C context). Top-right inset shows the multidimensional scaling (MDS) projection of this data. Each dot is a neuron colour-coded according to their cluster, and neurons with similar relative contributions occupy similar regions of the MDS plane. Source data are provided as a Source Data file.

30.6% of the fitted neurons were grouped around odour kernels combined with kernels related to visual context, modulations of odour responses by rewarded context, reward consumption and licking (Fig. 4e, 'odour' column). Since we use different first-session and expert animals in our experiments, we could not compare individual neurons before and after learning. Thus, neurons selective for odours before training could have acquired mixed selectivity after learning, or

a new associative neuronal population could have emerged that encodes several task-related variables in expert animals.

## Decoding of sensory information improves with mixed-selective neurons

We wonder whether the mixed-encoding has functional consequences, affecting the amount of sensory information carried by PCx responses.

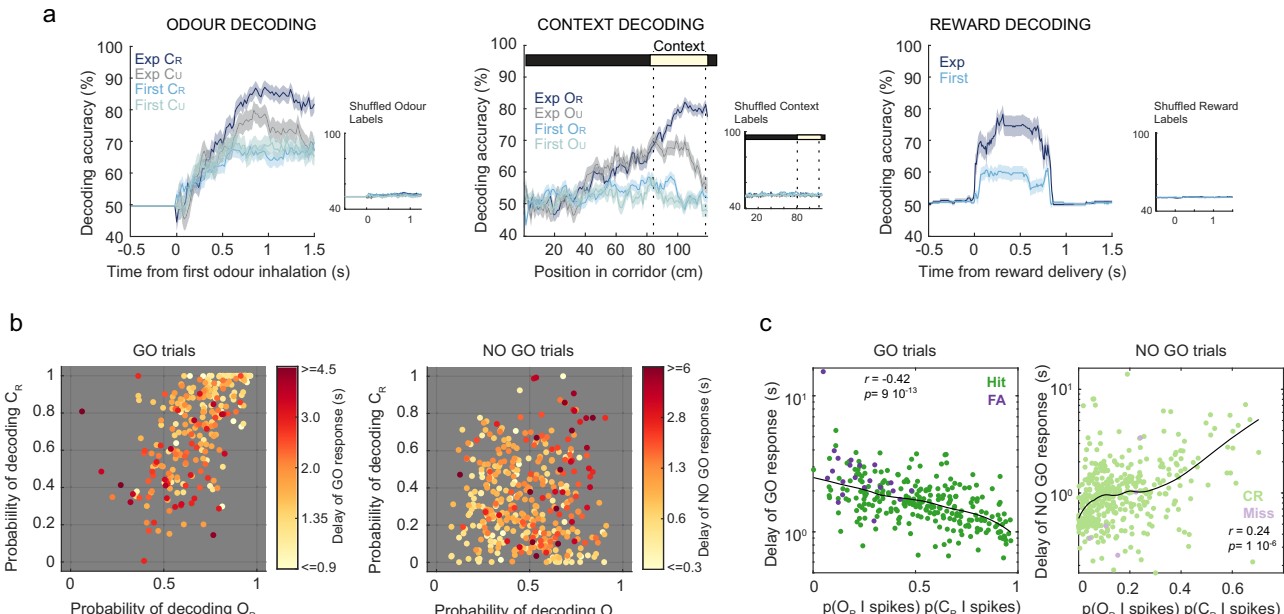

**Fig. 5 | Decoding of information from piriform neurons improves with learning and correlates with behavioural performance. a** GLM-based trial decoding accuracies (mean values ± SEM) of binary variable categories (rewarded vs. unrewarded odour, rewarded vs. unrewarded visual context, reward consumption vs. no reward consumption), using first-session and expert-session recordings. The number of trials used for model fitting and decoding analysis was matched across different trial types. **b** Delay of behavioural responses in GO trials and NO GO trials as a function of probability of decoding $O_R$, $p(O_R \mid \text{spikes})$, and probability of decoding $C_R$, $p(C_R \mid \text{spikes})$. In GO trials, the delay corresponds to the reaction time to first lick after first odour inhalation, while in NO GO trials it corresponds to the delay to leave the context zone after first odour inhalation. **c** Delay of behavioural response as a function of probability of the interaction term $p(O_R \mid \text{spikes})p(C_R \mid \text{spikes})$. Black curve is a smoothed moving average. Pearson correlation coefficient and two-sided $p$ value are indicated. Source data are provided as a Source Data file.

For that, we used the fitted GLM encoding models and the observed spiking activity of the PCx, to perform model-based trial decoding of odour, context and reward information (see 'Methods').

Interestingly, while the trial-by-trial activity of neurons in expert animals allows decoding the identity of odours and of visual contexts, the activity of neurons in first session animals only permits to decode information about odours (Fig. 5a). Decoding of contextual information was performed throughout the corridor, and accurate decoding of context information in expert animals only occurs in the proximity of the contextual zone as animals see the approaching visual cues in the distance (Fig. 5a). Trials with reward consumption are successfully decoded in both conditions (Fig. 5a). Importantly, PCx odour and context decoding correlated with behavioural performance in expert animals. First, we calculated the probability (given the observed PCx spikes) of decoding $O_R$ and $C_R$, $p(O_R|\text{spikes})$ and $p(C_R|\text{spikes})$, respectively, for each trial at 0.5 s after first odour inhalation (see 'Methods'). We then studied if the delay of the animal response (reaction time of licking in GO trials and time to exit the context zone in NO GO trials) was associated to these decoding probabilities (Fig. 5b). For that, we performed multiple linear regression of the delay of behavioural response against a model with 3 predictors consisting of $p(O_R|\text{spikes})$, $p(C_R|\text{spikes})$ and an interaction term $p(O_R|\text{spikes})p(C_R|\text{spikes})$, revealing that only the interaction term is significant for explaining the variability of response delays (F-test of overall significance for interaction terms: $p = 0.03$ for GO trials, $p = 0.02$ for NO GO trials). This interaction term quantifies the PCx representation of the rewarded combination. Indeed, behavioural responses in GO trials are faster when the representation of the rewarded combination is better (Fig. 5c, top panel). On the contrary, the behavioural response in NO GO trials was faster when the representation of the rewarded combination was worse (Fig. 5c, bottom panel). Overall, these results support that the degree to which the $O_R C_R$ association is successfully encoded in PCx translates into behavioural discrimination performance.

Notably, in expert animals, decoding of context in unrewarded odour trials decays to chance level once the odour was presented, while for the rewarded odour trials decoding performance is maintained at high levels (Fig. 5a). Thus, the learned contextual information is dynamically modulated along each trial, and persists in PCx only when context identity is needed to disambiguate reward outcome. This persistent activity is reminiscent of a working-memory trace that represents contextual information held in memory to determine trial performance, and further supports an associative function of PCx. In the same line, some of the associative neurons that we found show persistent excitatory or inhibitory activity in $O_R C_R$ trials (Associative neurons in Fig. 3c and Supplementary Fig. 3i, see neuron #3 N59).

Although odour identity could be extracted from PCx activity of both expert and first session animals, odour decoding in expert animals reached a higher accuracy and needed a fewer number of neurons than before learning (Fig. 5a and Supplementary Fig. 10a). Moreover, odour identity in expert animals was better decoded in trials when correct odour discrimination leads to a reward, that is, when the odour was presented in the rewarded visual context (Fig. 5a and Supplementary Fig. 10a). A similar result was obtained when decoding odour identity though dPCA population analysis, where decoding was faster and more accurate when odour was presented in $C_R$ trials compared to all trials regardless of context identity (Supplementary Fig. 10d). This indicates that the learned contextual modulation in the PCx enhances olfactory information during moments when odour discrimination becomes behaviourally significant.

If PCx non-olfactory modulation is indeed beneficial for odour decoding, we hypothesised that odour discrimination in PCx should be reduced when contextual modulation is absent, for example, when animals are passively exposed to the same odours. We recorded PCx neuronal responses during task engagement and, by subsequently turning off the virtual reality, the same neurons were recorded during passive exposure to both odours. Comparing neuronal activity under passive stimulation to that observed while performing the task

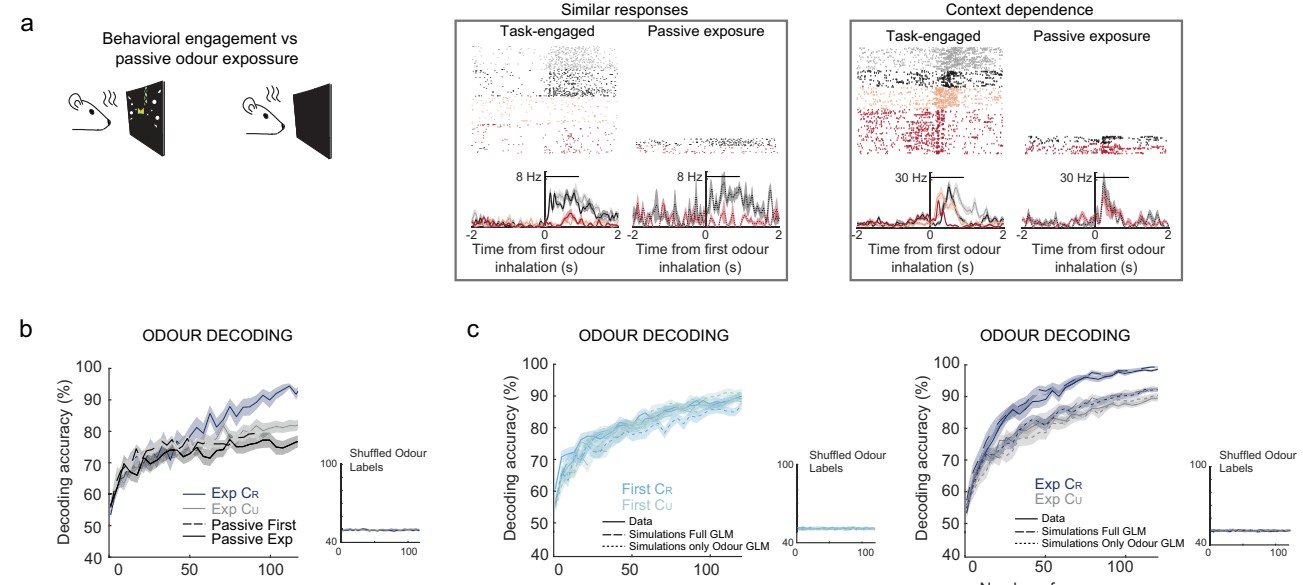

Fig. 6 | **Decoding improvement is dynamically engaged in PCx and depends on mixed selectivity. a** Left, the scheme represents the experimental condition, during task engagement or when the virtual reality was turn off and animals were passively exposed to odours. Right, examples of the responses of 2 neurons recorded in expert sessions during the task or under passive exposure to odours (mean values ± SEM). The first neuron has similar responses under both conditions, while the second neurons show context-dependence of the response. **b** Decoding accuracy (mean values ± SEM) of linear classifiers for odour identity as a function of neuronal population size, using expert-session data in $C_R$ and $C_U$ trials, and using neuronal responses from passive odour exposure. Decoding performed at the moment when accuracy first peaks (0.5 s and 1 s after first odour inhalation for task-engaged and passive conditions, respectively). Inset shows decoding accuracy after shuffling trial labels odour labels. **c** Decoding accuracy (mean values ± SEM) of linear classifiers for odour identity as a function of neuronal population size, for recorded data (Data) and simulations of GLM models including all fitted kernels (Full GLM) and after removing contributions of non-odour-related kernels from the simulations of odour-responsive neurons (Only Odour GLM). Left, accuracy for first-session data and simulations in $C_R$ and $C_U$ trials. Right, same as Top but for expert-session. Insets show results after shuffling odour labels. Decoding always performed at 0.5 s after first odour inhalation. For all decoding analysis shown in (**b** and **c**), the total number of trials used for training and testing classifiers was matched across the different trial types compared. Source data are provided as a Source Data file.

revealed that some neurons have similar responses in both conditions, while others showed dramatic changes in their responses (Fig. 6a and Supplementary Fig. 11). In particular, we found neurons with almost identical responses to both odours during passive stimulation, while the response of these neurons during task engagement allows discrimination of both odours and also of type of trial (Fig. 6a and Supplementary Fig. 11). Using linear decoders on the recorded neuronal populations (see 'Methods'), we observed a lower and slower performance of odour identity decoding under passive stimulation compared to the task-engaged condition (Fig. 6b). Furthermore, decoding accuracy of passively presented odours showed no difference between first-session and expert animals (Fig. 6b), ruling out a perceptual learning effect on the decoding accuracies observed in expert animals due to repeated odour exposure throughout training. These results suggest that at the population and even at the single neuronal level, the multimodal associative encoding acquired during learning leads to a faster and improved accuracy of odour identity decoding in the PCx.

To specifically test if the acquired mixed-selectivity underlies the improvement of odour identity decoding in rewarded contexts (Fig. 5a and Supplementary Fig. 10a), we used linear decoders on GLM simulations of PCx neuronal activity, which reproduced the odour decoding performance of the observed data (Fig. 6c). When we removed the contribution of non-olfactory kernels to the odour encoding neurons, odour decoding accuracy decreased in expert rewarded context trials, reproducing the decoding performance observed during the unrewarded context trials (Fig. 6c). We obtained similar results using GLM based decoding (Supplementary Fig. 10e). Interestingly, no single non-olfactory variable was responsible for the accuracy increase in rewarded context trials, and the effect was dependent on the presence of the multiple non-olfactory kernels of odour-coding neurons from expert animals (Supplementary Fig. 10f). These findings demonstrate that learned multidimensional mixed-selectivity allows PCx neurons to enhance odour decoding when odours acquire behavioural relevance.

## Discussion

Our findings show that as mice learn to recognise reward-predicting associations between olfactory stimuli and spatial contexts, the PCx can go beyond its role in odour identity encoding, showing a diversity of non-olfactory signals related to decision-making of learned behavioural responses. We found widespread contextual modulations of PCx activity which, when integrated with its canonical olfactory responses, allow ensembles of PCx neurons to better discriminate odours. The underlying computational mechanism is the acquisition of mixed-selectivity for diverse task-related variables by odour-responsive PCx neurons. This neuronal code is engaged in a dynamic manner depending on task demands, providing experience-dependent contextual and non-olfactory information to improve odour processing in the PCx and odour discrimination at the behavioural level.

Primary sensory cortices (PSCs) are known to encode features of stimuli from single sensory modalities. Nevertheless, prominent non-sensory responses have also been reported in PSCs. Pioneering studies found that locomotion can enhance the sensitivity gain of visual neurons in region V1 of the visual cortex[24–26], while it suppresses auditory responses in A1 of the auditory cortex[34,35]. Animal movements in general, and even task-unrelated ones, strongly modulate activity across the dorsal cortex, including V1 and primary somatosensory cortex S1[27,36]. Moreover, spatial maps have also been found in V1[22,37], and more recently in PCx[28], suggesting an unexpected function of PSCs in spatial cognition. The particular role of arousal and locomotion inputs to sensory cortices has been extensively debated and speculated

upon[15,38–41], but we still lack a description of how this variety of non-sensory inputs collectively interact with sensory responses at PSCs. It is not evident if the net effect of this interaction would facilitate or interfere with correct sensory processing. Here we found that, by combining olfactory inputs to the PCx with various behavioural and task-related signals that develop as a result of learning, odour decoding at PCx is enhanced. Remarkably, the enhancement is obtained when all signals are used in combination for odour decoding (Fig. 6c).

Encoding based on neuronal 'multitaskers' is often found in high-order prefrontal[42,43], parietal[44] and hippocampal[20] cortices. We show that learning shifts the mouse PCx cortex from the typical modality-specific sensory-driven functional organisation of primary sensory cortices, towards a mixed-selectivity associational regime. The dimensionality of odour-context responses increased following learning and was higher than the total number of odour-context combinations (Supplementary Fig. 2g). This, combined with the presence of odour-context interaction components (Fig. 2c and Supplementary Fig. 2b, d), indicates that PCx neuronal responses are a nonlinear mix of odour and contextual information. In association and executive brain regions, it has been suggested that nonlinear mixed-selectivity maps inputs into high-dimensional encoding spaces that allow simple downstream decoding operations to produce a wide range of context-dependent decision-making responses[16,17,45]. The learning-induced nonlinear mixed-selectivity observed in PCx can enhance odour decoding by similarly exploiting high dimensional encoding representations, which result from the combined action of multiple non-olfactory responses on PCx activity. These results indicate that the PCx could facilitate stimulus separability by reducing overlapping odour representations, promoting discrimination and avoiding generalisation of the same odour in different contexts, which is relevant for the correct performance of the task (Supplementary Fig. 12). It would be interesting to further investigate if these non-linear and linear encoding schemes could co-exist in PCx depending on animal state and learning level.

The difficulty of the task could play an important role in regarding the type of information that is represented in the PCx. The virtual environment allows the animals to operate on the task for the presentation of stimuli and rewards, allowing a different level of commitment and cognitive involvement than using simpler odour-reward associative tasks, in which reward signals were not observed in PCx[46,47]. Nevertheless, others have found that perceptual learning modulates the discriminability of complex odour mixtures in the activity of PCx[48,49] and associative and value information in PCx has been reported during odour-reward pairing[50–53]. Our results agree with these observations, further demonstrating the plasticity and versatility of PCx representations.

Our experiments indicate that the expert PCx exhibits responses to visual cues both before the presentation of an odour and during odour delivery (Fig. 2 and Supplementary Fig. 3i). These discoveries suggest the potential involvement of mechanisms related to anticipation triggered by visual cues as well as direct cross-modal influences on the piriform neurons, contributing to the improvement in odour processing. Expectation has been shown to influence the processing of sensory information in primary sensory cortices[54]. We ruled out the possibility of a generalised expectancy mechanism as no neurons responding to the visual context were observed in first session animals that are engaged in the task, and not all recorded piriform neurons show these modulations in experts. However, the odour enhancement observed in PCx in the rewarded visual context could involve memory-driven selective attention mechanisms triggered by the contextual cue that anticipates reward. Additionally, we found neurons that do not have anticipatory activity to odour delivery and solely respond to specific context-odour associations (Fig. 3c), even to non-rewarded context combinations (Supplementary Fig. 3i). This indicates that PCx can encode the identity of different cross-modal stimuli and that

modulation by visual context could also involve mechanisms other than reward expectation, such as direct cross-modal responses. These signals may arise by strengthening of PCx inputs from areas encoding reward expectation (e.g., orbitofrontal cortex[55] or basolateral amygdala[56]) acting in concert with other mechanisms involving the reinforcement of multimodal visual or positional/spatial inputs (e.g., coming from lateral entorhinal cortex or hippocampus[45]) after learning. It would be interesting to identify the sources of these diverse signals and clarify their role in shaping early sensory processing in PCx.

The present work provides insights into the plasticity of PCx encoding in response to experience. They highlight a computational mechanism at a primary sensory cortex that exploits the multiplexed nature of sensory experience to improve neural coding, by resorting to contextual information to dynamically amplify stimuli features when they are relevant for ongoing behaviour. We speculate that a similar process could take place in other primary sensory cortices known to be modulated by non-sensory information.

## Methods

### Animals

**Animals.** Mice were housed in a temperature-controlled room with an ambient temperature of $22° \pm 1°C$, $55 \pm 5\%$ humidity and a 12 h/12 h light/dark cycle. All experiments were performed during the dark cycle. All animal procedures were approved by the Animal Care and Use Committee at the IBioBA Institute (IBioBA-CICUAL # 2020-03-NE). We used 7–9 weeks old ($n = 10$ mice) female and male C57BL/6J mice (IBioBA Institute Facility). Six animals were pooled in the first-session group and four in the expert-session group. Blinding is not relevant to this study since all behavioural and neuronal activity analyses were performed with automated scripts without experimenter intervention or selection, and curation of spike-sorting results was done without knowledge of responsiveness to stimuli of each single-unit cluster. Littermates were randomly assigned to first-session or expert-session groups. No statistical methods were used to predetermine sample size. Mice were group-housed before head bar implantation and housed singly thereafter. The experimental timeline is shown in Supplementary Fig. 1a.

**Water restriction protocol.** Water restriction started after mice recovered from surgery (at least 3 days post-surgery, Supplementary Fig. 1a). Mice were individually housed in cages containing play tunnels, a wheel, and nesting material, in a reverse 12 h/12 h dark/light cycle housing room. The light in the accommodation room had an intensity of 100 lux at the height of the boxes, with white LEDs. The habituation, training and neuronal activity recording sessions were carried out during the dark phase. In the experimental device where the animals were placed, there was no illumination besides the light from the virtual reality screen. Dry food was continuously available. One ml of water was dispensed manually into small containers attached to the inner walls of the cages, always at the same time of day. Mice were monitored daily for hydration level, weight, skin health, and locomotion. To control these parameters, a daily record of: movement in the box, cleaning/grooming, posture, tension/relaxation of the skin of the neck, defecation in the housing box was used. Our protocol (IBioBA-CICUAL # 2020-03-NE) was subject to rigorous animal welfare control measures, including daily weight determinations and activity scores, posture, grooming, intake, and signs of dehydration. In the event of a weight loss below the minimum body weight (70% of the initial weight), the total daily volume of water they receive was increased until the stabilisation of the body weight in water restriction.

### Behavioural task

Mice were head-fixed and trained to run on a running wheel (a plastic cylinder) (Fig. 1a and Supplementary Fig. 1b). A water spout was positioned near the snout of the animal, and licks were detected with an

infrared beam and sensor. Wheel rotations as the animal ran were translated into displacements through a linear virtual corridor, which was presented in a screen in front of the animal. Mice were trained in a behavioural task of olfactory discrimination dependent on visual context. Here, visual context refers to the virtual environment, which could be one of two types of corridors. Each corridor consisted of an approaching aisle (83 cm in length), a context zone (33 cm in length) and a reward zone. Both types of corridors were visually distinct only in the context zone. The approaching aisle and the reward zone were similar for both corridors, with black walls displaying white spots. The context zone could be either green (green floor and walls with vertical white and green stripes, and a green column on the right with a black diamond-shaped pattern) or grey (black floor with white diamond-shaped pattern and grey walls with black dots, and a column on the left with horizontal white, black and grey stripes) (Supplementary Fig. 1b). When the animals entered the context zone, they received an odour puff of 1-s duration through a pipe directed to the animal nose. The odorant could be of two types: isoamyl acetate or ethyl butyrate. Once animals left the context zone where they received the odorant stimulus, they entered the reward zone, where they could choose to lick (GO response) or not (NO-GO response) to obtain a drop of water reward. There are four possible odour-context combinations, only one of these was rewarded after the GO response in the reward zone (Fig. 1b). No punishment was given for incorrect GO responses or for licking elsewhere in the corridor in any trial type. After reward delivery or an incorrect GO response the trial was immediately finished and a new trial started by 'teletransporting' mice to the start of the approaching aisle.

A specific rewarded odour-context combination (e.g., isoamyl acetate-grey context) was randomly assigned to each mouse and maintained throughout all training sessions. This rewarded combination was labelled $O_RC_R$. The individual odorant and context used in that rewarded combination were labelled $O_R$ and $C_R$ respectively, while the remaining odorant and context were labelled $O_U$ and $C_U$. The 'R' and 'U' subscripts stand for 'rewarded' and 'unrewarded', but notice this is an abuse of language since specific odour-context combinations were rewarded, not individual odours or contexts (that is, only $O_RC_R$ trials were rewarded while $O_RC_U$ and $O_UC_R$ trials were not).

**Passive stimulation.** For four of the first-session and four of the expert animals, at the end of the behavioural task we additionally performed a passive odorant stimulation protocol. The protocol consisted of 20 trials of stimulation with each odorant (trials were randomly interleaved), while the virtual reality monitor was turned off (Fig. 6a). Odours pulses lasted 1 s, and were presented every 40 s.

**Experimental apparatus.** The device consisted of: (1) Columns for holding the animal's head (head-fixed rig). Animals are fixed to the rig by clamping a skull-implanted metal bar (headbar) to the rig's columns. (2) A running wheel for the animal with a rotary encoder attached to the wheel's axle that records wheel rotations as the animal walks (see 'Data acquisition'). The wheel is made of plastic and is covered with black EVA rubber so that the substrate is soft to the touch of the animal and avoids slippage; (3) A circuit of valves connected to pipes (olfactometer) that conducts the odour to the animal's nose (see 'Olfactometer' in 'Odour stimuli'). This system is computer-controlled, which allows different odours to be presented with adjustable flow rates and to switch between odours quickly (<20 ms). In addition, an air exhaust quickly removes the odour after its presentation. Both the olfactometer and the exhaust allow the olfactory stimulus to be presented in a timely manner. Airflow is kept constant and regulated by flowmeters; (4) An inhalation-exhalation cycle measurement system: sniffing behaviour was recorded with a mass airflow sensor located externally in close proximity to the animal's left nostril (see 'Data acquisition' and 'Analysis of respiration recordings'). Precise

orientation relative to the nostril was manually optimised before each recording to attempt to acquire a full signal, despite any movement of the nose. (5) A custom-made lickometer for measuring the animal's licking response and for delivery of water rewards (see 'Data acquisition'). (6) A computer screen placed in front of the animal that displayed the virtual reality environment according to the task protocol. (7) A microcontroller and data acquisition system (Bpod State Machine r1, Sanworks) that records the animal's licking response and its behavioural performance across trials, while controlling in real time the state of the task (activation of odour valves, delivery of reward, sequence of trial types) according to the task protocol. (8) A data acquisition system (Smartbox, Neuronexus) that acquires and stores behavioural and trial information, along with the electrophysiological signals (see 'Data acquisition'). (9) A CPU running MATLAB (Mathworks) for controlling the BPod and running the Virtual Reality MATLAB Engine, ViRMEn[57].

## Animal training

Behavioural training: after body weight stabilisation, usually after 5 days of water restriction, the experimental device habituation and training session began (Supplementary Fig. 1a). During those days, we carried out a daily *handling* or manipulation session in which the animals got used to the manipulation of the experimenter and to remain comfortable in the hands of the experimenter. Habituation to the experimental device: 3 days before training, the mice were manipulated daily so that they got used to the environment of the training room and head fixation to the experimental device. Behavioural task training: a random sequence of trial types (different odour-context combinations) was presented to the animal in each training session. Individual training sessions lasted 40–60 min. In the first training session animals completed $134 \pm 51$ trials, while they performed $227 \pm 60$ trials on expert sessions. Rewards were 10 microliter water drops.

## Odour stimuli

**Olfactometer.** Odours were delivered using a custom made olfactometer. Charcoal filtered air was routed into two flowmeters (03216-06 and 03216-16, Cole-Parmer, IL). The first one carried a neutral air stream at 0.9 liters per minute (LPM) that was kept constant (base stream). The second one carried a 0.2 LPM stream which was further splitted in two 0.1 LPM arms with an injection valve (SI360T041, NResearch, NJ; 'odour bank valve'). When the valve was turned off the first arm was selected and the air stream traversed through a 10 ml empty vial containing (blank stream). When the valve was turned on the second arm was selected and the air stream was channelled to a second injection valve ('odour selection valve') that routed the air to one of two 10 ml vials, each one containing a 2 ml solution of one of the two odours. The voltage state of this odour selection valve thus controlled which odour would compose an odorant stream. The two arms were combined in a shuttle valve (SH360T041, NResearch, NJ; 'isolation valve') that was synchronously activated with the odour selection valve to isolate both odour paths, avoiding odorant cross-contamination that could result from air reflux into the remaining vial. The odorant stream and the blank stream were then fed to a final shuttle valve that selected between both streams ('injection valve'). The selected 0.1 LPM stream (odorant or blank) was combined with the 0.9 LPM base stream to produce a 1 LPM stream that was routed to the animal nose. The Bpod system was programmed to control valve voltages so that when entering the visual context, the appropriate odorant stream would be selected and the animal would receive the 1-s-long odour pulse scheduled for that trial. Final valve switching simultaneously rerouted the odorant stream to the animal's nose and the blank stream to a vacuum exhaust, and switched back after 1 s. At any other trial moment, the blank stream would be selected and the animal would receive an air stream that passed through vials containing only

mineral oil. Non-selected odorant streams were also directed to the vacuum exhaust. Consistency of shape and arrival times of odour pulses were monitored with a custom-made photoionization detector (PID; PID-A1, Alphasense, Essex, UK) located at the outlet of the stimulation tubing near the animal's nose. Calibration of the olfactometer was routinely performed such that switching of the injection valve produced minimal perturbations of the total air stream to the animal's nose. Tubings were made of Teflon to avoid accumulation of residual odours.

**Odorant stimuli.** We used isoamyl acetate or ethyl butyrate (SIGMA-Aldrich). The odours were chosen based on literature showing that these two odours had no innate response in mice[6]. The odours mentioned were prepared in the liquid phase at room temperature and the volatiles of the gaseous phase were supplied to the animals as a stimulus. For all experiments, solutions with 1:200 dilutions were used (odour: mineral oil) for odorant vials. The volatiles present in the gas phase of the vials were administered to the animals through the olfactometer. Odorant and blank vials in the olfactometer contained a liquid phase of 2000 µl of the respective solutions and a gas phase of 10 ml in which vapours were accumulated until reaching equilibrium with their respective vapour pressure. As a source of purified air, an aquarium pump was used whose air was passed through activated carbon and cotton filters. Airflow was constantly controlled by valves driven by electrical controllers. The opening and closing of the air flow passing through each vial was regulated. When a valve was opened, a volume of the gaseous portion of the vial was displaced, mixed with a continuous main flow of purified air, and finally emptied into a tube with an outlet located 1–2 cm from the mouse's head.

## Surgeries
**Stereotactic targeting and head bar attachment surgery.** Animals were injected intraperitoneally with Ketamine (100 mg/kg body weight) and Xylazine (10 mg/kg body weight). From the establishment of total anaesthesia, controlled through the loss of body reflexes, the operation lasted a maximum of 30 min. It was done on a heating pad to preserve the animal's body heat. The eyes were protected from drying out by applying ophthalmic gels such as Vidisic (active ingredient: polyacrylic acid). The entire procedure was performed under aseptic conditions. Once anaesthetised, animals were placed in a stereotaxic. A dose of bupivacaine 0.5% (50 µl) was injected under the skin as a local anaesthetic, and we waited for 5 min before continuing. An incision was made in the skin to expose the skull, disinfecting the area with pervinox using sterile swabs. The animal was prepared to place a metal bar that allowed us to attach its head to the experimental device and keep it fixed, and on the other hand to mark the skull in the position where the recording electrode would be inserted afterwards. The scalp was cut, exposing the skull and the region to be drilled. The skull was cleaned and dried with sterile cotton swabs for better adhesion of the glue. The 2.5 cm × 0.5 cm, 350 mg weight aluminium headbar was placed directly on the wet glue and then dental acrylic was added to cover the glue and cement the bar in the desired position. The metal bar then stuck rigidly to the skull, without going through it. The bar was attached to the apparatus during behaviour. With the aid of a mouse brain atlas, the desired position relative to the bregma in the left hemisphere (for piriform cortex AP 3.2 mm; ML 3 mm) was reached using a stereotaxic and a point was marked. We also implanted the ground and the reference electrodes (silver wires) in the cerebellum. After the operation, animals recovered on the heating pad until awakening from anaesthesia. We have determined that within 2 h after the operation mice wake up, walk, drink, eat and do not show behaviour indicative of pain sensation. However, preventively, the analgesic Tramadol was administered in a subcutaneous dose (5 mg/kg body weight), and in the drinking water. In addition, an anti-inflammatory,

Ketoprofen, was administered by subcutaneous injection (5 mg/kg body weight) one dose, for 2 days. Animals were allowed to recover from surgery for 3 days.

## Neural recordings
In different training sessions, depending on the experimental group to which the animal belongs (first session or expert animals), the recording session was held, which consisted of presenting a sequence of different trials while recording the activity of neurons in the piriform cortex. To do this, 1 day before this session, the craniotomy was performed where the recording electrode was going to be acutely inserted the following day. For this, animals were anaesthetised with isoflurane (2% induction, 0.5–1% maintenance). A drill was used to gently file down the bone in the marked area during the previous surgery. Then, with the help of a needle, a small 'cap' was lifted, exposing a small area of the brain which was kept moist carefully using swabs moistened with saline solution, avoiding touching the brain. Finally, the exposed portion was covered with a thin layer of cyanoacrylic glue (WPI). At the end, 100 µl of an anti-inflammatory, Ketoprofen, was administered by subcutaneous injection (5 mg/kg body weight), and then animals were allowed to wake up and recover. The next day, the awake animal was placed in the experimental device, the glue/gel covering the craniotomy was removed with forceps, and an array of recording micro-electrodes (Silicon probes, Neuronexus) attached to a micromanipulator was inserted. With the aid of the micromanipulator we reached the piriform cortex, descending slowly (1 micron/second) to the target area (DV 3.8–5.2 mm, depending on the animal). Once we arrived, we waited 20 min to stabilise the recording position. The recording probe was previously painted with a dye (DiI D3911, Thermo Fisher). All recordings were performed using A1x32-Poly3-5mm-25 s-177 silicon probes (177 µm² site surface area, 3-column honeycomb site geometry with 18 µm lateral and 25 µm vertical site spacing, 36 µm centre-to-centre horizontal span, 275 µm centre-to-centre vertical span, 114 µm maximum shank width near the sites, 15 µm shank thickness) with an H32 connector (NeuroNexus Technologies). The animal performed the behavioural task while the neuronal activity was recorded. Once the recording session was complete, the animal was placed in its housing cage with water ad libitum. The next day, it was anaesthetised intraperitoneally with Ketamine (100 mg/kg body weight) and Xylazine (10 mg/kg body weight). Once anaesthetised, without reflexes and in an unconscious state, the animal was decapitated and the brain dissected. The animal's brain was used for verification of the location of the recording probe (Fig. 3a). To do this, the brain was cut into 50-micron slices on the cryostat, the slices were stained in DAPI preparations to visualise cell nuclei under a fluorescence microscope Axio Observer (Zeiss), and the location of the probe tip was revealed by the fluorescence of the dye DiI. Images were analysed with Zen Zeiss pro 2011 (Zeiss).

## Data acquisition
Electrophysiological signals were acquired with a 32-site polytrode acute probe (A1 × 32-Poly3-5mm-25s-177, Neuronexus, MI) connected to an Acute Smartlink32 headstage (Neuronexus, MI). Unfiltered signals were digitised at 30 kHz at the headstage and recorded by a Smartbox multichannel data acquisition system (Neuronexus, MI). Experimental events and respiration signals were acquired at 30 kHz by analogue and digital inputs of the Smartbox system using Allego Smartbox versión 0.7.3.0-20180628 software (Neuronexus, MI). Respiration was monitored with a microbridge mass airflow sensor (Honeywell AWM3100V, NJ) positioned directly opposite the animal's nose. Negative airflow corresponds to inhalation and negative changes in the voltage of the sensor output.

Behaviour was automatically monitored by a behavioural measurement system Bpod SanWorks Console v1.77 (Sanworks, NY), which was programmed to control task flow according to the protocol (see

'Behavioural task') and sent behavioural timestamps using BNC signals to the Smartbox system for synchronisation with neuronal signals.

Animal displacement on the running wheel was measured with an optical rotary encoder (H5-360-IE-S, US Digital, WA) attached to the wheel axle. When the animal moved the encoder sent TTL pulses to the Smartbox (for posterior synchronisation with neuronal signals), and to a microcontroller (MEGA 2560, Arduino, NY) programmed to calculate animal position. The microcontroller transmitted the positional information to the Virtual Reality MATLAB Engine (ViRMEn). It also and TTL pulses to the Bpod to signal context zone entry (to trigger odour stimulation) and exit (to signal entry to reward zone).

## Data analysis and statistics
All data analysis and statistical tests were performed with custom-written software using MATLAB (Mathworks). Statistical significance was assessed with the Wilcoxon rank sum test, unless otherwise noted.

## Analysis of task performance
Task performance of the animals was quantified using Signal Detection Theory, SDT[58,59]. Briefly, through analysis of the Hit rate and False-Alarm rate of a subject performing a discrimination task, SDT allows to disambiguate the contributions of the perceptual strength of the stimulus and the subject's bias to respond.

The parameter called $d'$ (also *d-prime* or sensitivity) measures the perceptual strength of the target stimulus, that is the discriminability between the subject's perceptual representations associated to the presence or absence of the target stimulus:

$$d' = Z(Hit\,\text{rate}) - Z(FA\,rate) \tag{1}$$

where $Z$ is the normal inverse cumulative distribution function, and $Hit_{rate}$ and $FA_{rate}$ are the Hit and False-alarm rates, respectively. In our task, the target stimuli are the $O_R C_R$ trials. A value $d' = 0$ is chance performance, and higher values of $d'$ indicate better performances. For example, for $Hit_{rate} = 95\%$ and $FA_{rate} = 5\%$ one obtains $d' = 3.29$ (dotted line in Fig. 1c).

The parameter called *criterion* (also *c*) is related to the subject's bias response:

$$criterion = -0.5 \times [Z(Hit\,\text{rate}) + Z(FA\,rate)] \tag{2}$$

According to SDT, the subject decides to respond or not in a given trial by imposing a threshold: if the perceptual representation in the trial is larger than the threshold *criterion* then the subject responds with a 'GO'. Negative criterion values indicate a bias towards *GO* responses, positive values indicate a bias towards *NO-GO* responses, and *criterion* = 0 indicates no bias (neutral response).

In our task, initial animal training sessions had negative *criterion* and low $d'$ values, indicating almost chance performance and a bias towards licking across all trials (since animals were thirsty and there was no punishment for False Alarms). As training progressed, animals steadily increased $d'$ (increasing performance) and brought *criterion* towards a neutral unbiased response (*criterion* $\longrightarrow$ 0).

## Processing of respiration recordings
We took an approach similar to previous studies[60]. Respiration traces sampled at 30 kHz along with the neural recording signals were smoothed with a second-order Savitzky-Golay filter in 100 ms frames and the result was locally detrended by subtracting its 1-s-long median-filtered signal. The start of inhalation was defined as zero-crossings before large negative peaks in the smoothed, detrended signal. Odour inhalations were defined as inhalations happening during a 1-s long time window starting immediately after the arrival of the odorant pulse.

## Spike sorting
Extracellular voltage traces were preprocessed with common median referencing (subtraction of the median across all channels at each time sample to remove artifacts), spike sorted using Kilosort[61] (https://github.com/cortex-lab/Kilosort) and the obtained result was manually curated with the phy GUI (https://github.com/kwikteam/phy). During manual curation all clusters of putative spike events detected by each template were inspected and evaluated according to a number of criteria. First, we discarded clusters that were considered noise if their events had near-zero amplitude or if event waveforms were non-physiological and/or extended across all recording channels. Clusters containing inconsistent waveform shapes or large number of refractory period violations (<2 ms) in the autocorrelogram were also discarded. In a final step we merged pairs of clusters that had similar waveforms, showed refractory period cross-synchronisation or temporally coordinated cross-correlograms that indicated a bursting neuron. Units that passed these criteria were labelled as single units and considered in our study.

## Single-neuron responses
For plotting and analysis of neuronal responses we used peri-stimulus time histograms (PSTHs) that were temporally smoothed with a gaussian filter (s.d., 30 ms). These 'event PSTHs' described the neuronal spiking rate temporal evolution around the timing of either the onset of sensory stimuli, task events or behavioural events (examples in Fig. 3 and Supplementary Fig. 3). For describing average responses along a complete trial we constructed PSTHs that were obtained by stitching together the 'event PSTHs' corresponding to the following sequence of trial events: visual context entry, first odour inhalation, animal's response, and trial outcome (labelled as 'C', 'O', 'L' and 'R' in Figs. 1e, f, 2c and 3d). The 'animal's response' trial event refers either to the timing of the first animal lick after odour delivery (in GO trials) or to the timing at which the animal leaves the visual context (in NO-GO trials). The 'trial outcome' trial event refers to the timing of reward delivery (in HIT trials) or to the timing that corresponded to an additional median inter-lick time interval after the 'animal's response' trial event (in MISS, FALSE ALARM and CORRECT REJECTION trials; the rationale for this is that rewards were delivered in the second lick performed in the reward zone). Each one of these 4 'event PSTHs' (C, O, L and R) were aligned to each event median time across trials, relative to first odour inhalation (for first session data C, O, L and R were aligned to −0.46, 0, 2.45 and 2.99 s, respectively; for expert session data C, O, L and R were aligned to −0.43, 0, 1.38 and 1.96 s, respectively). The final PSTH that described average activity of a neuron along a full trial was obtained by stitching together this sequence of time-aligned 'event PSTHs' by averaging periods where there was temporal overlap between them. We confirmed that this stitching procedure did not distort the shape of the individual 'event PSTHs' and precisely described the sequence of firing rate variations along the trial. All stitched PSTHs are shown in Fig. 1e, f and Supplementary Fig. 8 and (where the mean firing rate of each neuron was subtracted to display variations in firing rate around the mean).

## Principal component analysis (PCA) and demixed PCA (dPCA)
Dimensionality reduction of the population activity data was performed with PCA or dPCA[29]. The dPCA analysis was performed with the Matlab implementation provided by the authors (http://github.com/machenslab/dPCA). Details on dPCA can be found in the original publication, here we give a brief outline and describe how we applied dPCA to our data.

The data were organised by pooling all the full-trial stitched instantaneous firing rates across recordings and stimulus conditions, resulting in one array $X$ for each training stage (i.e., first-session or expert-session stage). The dimensions of $X$ were $N \times C \times O \times T \times K$, where $N$ is the total number of neurons across recordings, $C = 2$ is the number of contexts, $O = 2$ is the number of odours, $T$ is the number of

firing rate time samples, and $K$ is then number of trials. Thus, for each training stage, the array $X$ has $N$ rows in which the $i$-th row contains the instantaneous firing rate of the $i$th neuron for all stimulus conditions and all trials (the firing rates were centred, i.e., with row means subtracted). By averaging $X$ over all $K$ trials for each neuron, odour, and context, we obtained a matrix $\bar{X}$ of dimensions $N \times C \times O \times T$ collecting the centred full-trial stitched neuronal PSTHs.

For PCA, the 4-dimensional array $\bar{X}$ was reshaped into a matrix $\bar{X}_{reshape}$ with $N$ rows and $M$ columns, where $M = C \cdot O \cdot T$ (that is, concatenating all stitched PSTHs conditioned on odour-context combinations), and singular value decomposition was applied on $\bar{X}_{reshape}$ to obtain the matrix $W$ of principal components. Trajectories through principal component space were obtained performing the projection $\bar{X}^T_{reshape} * W$, where $^T$ refers to the transpose, reshaping this projection back to the 4-dimensional $N \times C \times O \times T$ form and plotting only the first 3 rows that correspond to trajectories through the top 3 principal components (Fig. 2b).

The aim of dPCA is to reduce the dimensionality of the data, while obtaining latent components that 'demix' task parameters. For this, the single-trial data array $X$ was reshaped into a $X_{reshape}$ with $N$ rows and $P$ columns, where $P = C \cdot O \cdot T \cdot K$. When shaped this way, $X_{reshape}$ can be linearly decomposed into a set of 4 marginalisations $x_\Phi$: one condition-independent term, and 3 additional terms containing firing rate variations specifically related to either odorant identity, visual context identity, or the interaction between odorant and visual context identities. Each of these marginalisations $x_\Phi$ can be understood as class means of the data, associated with the task parameter $\Phi$. Importantly, these terms are all uncorrelated, thus the covariance matrix $X_{reshape}X^T_{reshape}$ can also be linearly decomposed into the sum of covariance matrices $x_\Phi x_\Phi^T$ corresponding to each of the 4 individual marginalisations. dPCA thus seeks to find separate encoding and decoding matrices that, when applied to $X_{reshape}$, they minimise the least-squares reconstruction error of each individual $x_\Phi$ simultaneously. It can be shown that this minimisation is a reduced-rank regression problem that can be analytically solved by SVD (hence its relation to PCA). This dimensionality reduction step allows to infer a few latent components, also referred as decoding axes, that 'demix' each task parameter and can be used as linear classifiers to properly decode the corresponding parameter throughout time in single trials (Fig. 2c and Supplementary Fig. 2). Cross-validation was used to measure the time-dependent trial classification accuracy, and the significance of this accuracy was assessed by comparing performance against a shuffled set of trials (black horizontal bars in Fig. 2c and Supplementary Fig. 2c, d indicate periods of significant classification accuracy). To avoid overfitting with dPCA we included the regularisation term controlled by a parameter chosen through cross-validation on each dataset (results obtained without regularisation were qualitatively similar).

In order to decompose data over a specific set of parameters, dPCA requires the data to contain trials for all possible parameter combinations and all neurons. Therefore, we could not further decompose our data over a decision (GO / NO-GO) parameter, since some parameter combinations were not available in the recordings (e.g., across expert-session recordings only one animal produced NO-GO responses in $O_R C_R$ trials and that same animal did not produce GO responses in $O_U C_U$ trials). Nevertheless, clear decision-related information could still be observed in our dPCA data decomposition, particularly in expert recordings, as departures of $O_R C_R$ trajectories from the rest of the conditions around the animal response time. This was possible since 90.3% of GO responses (243 out of 269) were concentrated in $O_R C_R$ trials when animals become experts.

### Estimation of signal variance and dimensionality of odour responses

To estimate the fraction of total signal variance in the recorded PSTHs (dashed lines in Supplementary Fig. 2a, c, f), we followed the approach

described by Kobak et al. in the dPCA formulation (Kobak[29]). The rationale is that, due to the finite amount of recorded trials, the estimate $\bar{X}$ must differ from the 'true' underlying PSTHs, and the total variance in $\bar{X}$ must contain contributions from residual noise in the trial-averaged firing estimates.

The total variance in $\bar{X}$ can be calculated as:

$$\left|\left|\bar{X}\right|\right|^2 = \sum_{n,o,c,t}^{N,O,C,T} \left( \overline{X_{n,o,c,t}} - \left\langle \overline{X_{n,o,c,t}} \right\rangle_{o,c,t} \right)^2 \tag{3}$$

where n, o, c, t are subindexes for neuron number, odour type, context type and time sample, respectively, and angle brackets indicate averaging over the corresponding variables.

The estimate of the average noise variance across conditions for neuron n is related to the sum of the square differences between firing rates in each trial and the trial averaged firing rates:

$$\widetilde{C_n} = \sum_{k,o,c,t}^{K,O,C,T} \frac{\left( X_{n,o,c,t,k} - \overline{X_{n,o,c,t}} \right)^2}{trialNum_{n,o,c}} \tag{4}$$

where $k$ is a subindex for trial number, and trialNum indicates the total number of trials in each odour and context condition.

Thus, the total residual noise sum of squares is:

$$\Theta = OCT \sum_n^N \frac{\widetilde{C_n}}{\overline{trialNum_n}} \tag{5}$$

Where:

$$\overline{trialNum_n} = \frac{1}{OC} \sum_{o,c}^{O,C} trialNum_{n,o,c} \tag{6}$$

is a rebalancing term for pooling together sequentially recorded neurons with different total number of trials between conditions.

Finally, the total signal variance is:

$$SigVar = ||\bar{X}||^2 - \Theta \tag{7}$$

and the dashed lines in Supplementary Fig. 2a, c, g report the fraction of total signal variance, which is $100(1 - \Theta/||\bar{X}||^2)$.

To characterise the dimensionality of odour responses we estimated the total number of principal components needed to explain the fraction of total signal variance (Supplementary Fig. 2g). The fraction of total signal variance in the population of responses aligned to odour stimulation was higher for expert animals, indicating that odour responses are more reliable after learning. This was true even when matching the total number of trials in both training states: 71.7% and 55.4% total signal variance for experts and first sessions, respectively (dashed lines in Supplementary Fig. 2g). To confirm the increase in reliability in the population of odour responses after learning, we calculated the coefficient of variation to estimate trial-to-trial variability in PCx neuronal firing rates. Odour stimulation reduced the average coefficient of variation, as previously reported in rats[5,28], but the reduction was stronger for expert animals (coefficient of variation of spike counts across trials: $1.70 \pm 0.05$ for first sessions and $1.8 \pm 0.04$ for experts, mean ± s.e, $p = 0.02$, Supplementary Fig. 2f). This supports the lower total signal variance of odour responses observed in first session animals.

### Generalized linear model (GLM) for encoding and decoding analysis

For this analysis the spiking-activity time series were discretized into 10-ms bins. The analysis was performed by adapting the Matlab

implementation of Poisson GLM regression for trial-based spiking data by Park et al.[33] (https://github.com/pillowlab/neuroGLM).

This framework poses an encoding model $p(\mathbf{r}|\mathbf{x})$ of the probabilistic relationship between a neuronal response $\mathbf{r}$ and a set of task variables $\mathbf{x}$ in a given trial. The probability of response is related to an underlying time-varying spiking rate $\lambda_t$, which is given by:

$$\lambda_t = exp\left(\sum_i (k_i \otimes x_i)_t\right) \tag{8}$$

where $x_i(t)$ is the time course of $i$th variable in the model, $k_i$ is the neuron's 'kernel' (i.e., linear filter) associated with that variable, and $\otimes$ indicates a linear convolution operation. The kernel captures the relationship between this variable and the neuron's probability of spiking. In turn, in a given single trial, the probability of spiking of the neuron is given by a Poisson distribution (i.e., it assumes Poissonian spiking statistics):

$$p(\mathbf{r}|\mathbf{x},\{k_i\}) = \prod_{t=0}^{T} p(r_t|\mathbf{x},\{k_i\}) = \frac{1}{r_t!}\prod_{t=0}^{T}(\Delta\lambda_t)^{r_t}e^{-\Delta\lambda_t} \tag{9}$$

where $\Delta$ is the time bin size (10 ms), $T$ is the total number of time bins in the trial, and $r_t$ is the spike count at time bin $(t, t+\Delta)$.

**Parametrization of model variables and kernels.** A schematic representation of the parametrization is shown in Supplementary Fig. 5a. This choice of regression variables was based on the neuronal response modulations observed in the PSTHs and in population dynamics (dPCA). In our model there were two kinds of kernels: 'event-based' kernels (i.e., kernels reflecting the time-varying influence of a task variable on the time-varying spike rate), and 'tuning-based' kernels (i.e., kernels describing the dependence of neuronal firing on the value of a task variable). Kernels were parametrised by a set of basic functions (Supplementary Fig. 5b). For any kernel parametrised by $n$ bases, the model infers a set of $n$ coefficients that correspond to the weighted linear combination of the $n$ basis functions. To choose the number of bases $n$ used to parametrise each kernel, we ran a series of preliminary tests on a set of neurons and evaluated parametrization performance through cross-validation (see 'Model fitting and model performance').

**Event-based kernels.** Each event associated with the variable was represented as a delta function over time, and convolved with the corresponding kernel (Supplementary Fig. 5a). To parametrise the associated event kernels, we used basis functions defined by a series of raised cosine bumps centred at different times spanning the range of time covered by the kernel (Supplementary Fig. 5b):

- Inhalation onset: 10 bases covering 460 ms window after the onset of each inhalation. Labelled 'Inhal'.
- Inhalations of rewarded and unrewarded odours: 10 bases covering 460 ms window after the onset of each odour inhalation during odour stimulation. One kernel for each odour, labelled '$O_R$' and '$O_U$'.
- Licking: 15 bases covering 360 ms window after the onset of each lick. Labelled 'Licks'.
- Reward consumption: 15 bases covering 360 ms window after the onset of each lick. Labelled 'Reward'.
- Anticipation to GO response: 22 bases covering 2000 ms before the first lick that follows odorant stimulation. Labelled 'preGO'.
- Modulation of odour responses by presence of rewarded context: 20 bases covering 1500 ms after the first odour inhalation. One kernel for each odour, labelled 'modO_R' and 'modO_U'. These kernels aim at describing the interaction between odours and contexts.

**Tuning-based kernels.** Each variable was discretized and represented as an animal-state vector for the variable at a given instant in time, and

convolved with the corresponding kernel (Fig. 5a). Each animal-state vector denotes a binned variable value, all of whose elements are set to 0, except for one element, which is set to 1, corresponding to the bin the current animal state occupies. To parametrise these tuning kernels, we used basis functions defined by a series of raised cosine bumps centred at different variable values along the range spanned by the animal-state vector (Supplementary Fig. 5b):

- Animal position along rewarded and unrewarded visual contexts: 42 bases covering the entire length of the virtual corridor discretized in a sequence of 4-cm bins. One kernel for each context, labelled '$C_R$' and '$C_U$'.
- Running speed: a sequence of bins of 1 cm/s up to the animal's maximum running speed (the total number of bases varied between animals depending on their maximum speed, typically ~35 bases were used). Labelled 'Speed'.

**Bias kernel.** To account for the average firing rate of the neuron during the experiment, we introduced a bias kernel parametrised by a single coefficient that was convolved with a boxcar function that lasted for the whole recording (that is, the bias kernel was constantly active). This kernel is not shown on Supplementary Fig. 5a.

**Model fitting and model performance.** To fit the weights of the basic functions that parametrise the kernels in the GLM encoding model of each neuron, we maximised the model's log-posterior, using a ridge prior to regularise the inferred weights (under the assumption that kernels should be relatively smooth):

$$L(\mathbf{k}) = \sum_{t=0}^{T}(r_t\log(\lambda_t\Delta) - \lambda_t\Delta) + C - \xi||\mathbf{k}||^2 \tag{10}$$

where $\mathbf{k}$ is a vector representing the weights to be obtained, $C$ is a constant that does not depend on $\mathbf{k}$ (thus irrelevant for the optimisation procedure), and $\xi$ is the smoothing hyperparameter controlling regularisation.

Model performance was quantified through cross-validation, by computing the model log-likelihood (the log-posterior without the last regularising prior term) of held out data under the model. For this, we randomly divided trials into 10 sets of equal size. The cross-validation procedure was repeated on each of these 10 sets of trials (10-fold cross-validation; the log-likelihood of the model on each fold was normalised by the number of spikes in the fold). This procedure penalised model overfitting, thus allowing valid performance comparisons between models of varying complexity.

To select the hyperparameter $\xi$ we took an Empirical Bayes approach, in which we maximised the marginal likelihood (also called model evidence) with respect to $\xi$ (chosen from a grid) and used this estimate of $\xi$ into the posterior. To compute the marginal likelihood we used the Laplace approximation. This procedure does not require cross-validation, and it was performed on all trials at once.

Once a model was selected (see 'Model selection'), the final estimate of $\mathbf{k}$ was inferred by fitting the model on all trials at once.

**Model selection.** Three variables were associated with 2 kernels that depended on the identities of odorants and/or contexts: inhalations of rewarded and unrewarded odours ($O_R$ and $O_U$), modulation of odour responses by presence of rewarded context (modO_R and modO_U), and position along rewarded and unrewarded visual contexts ($C_R$ and $C_U$). When regressing neuronal responses against these 3 variables, the 2 corresponding kernels were used. Thus, the total number of regression variables was 8, and the models considered included combinations between all of them.

Testing all combinations of variables is intractable. Thus, we implemented a forward-search model selection procedure similar to the one introduced by Hardcastle et al.[62], that selects the simplest

model that best predicts held out neuronal activity data (spike-normalised model log-likelihood on 10 cross-validation data folds; see 'Model fitting and model performance'). First, models including a single variable were evaluated, and the one with highest performance was determined. This single-variable model was then compared against all models with two variables that included this single variable. If the double-variable model with highest performance was significantly better than the best single-variable model, we proceeded to compare this double-variable model to the best performing triple-variable model that included this pair of variables. We continued including variables in the model through this procedure until the more complex model considered did not show an improvement in prediction performance, in which case the simpler model was selected. Significance in performance improvement was always assessed with a two-sample one-sided signed rank test, which tests the hypothesis that the difference in performance across data folds between the more complex and the simpler model comes from a distribution with median greater than 0. Models were considered better performing if they had a $p$-value lower than 0.05 under this test. Neurons for which the selected model did not perform significantly better than a model with constant mean firing rate were considered as not modulated by task variables (i.e., they had 0 modulating variables).

**Validating the exponential nonlinearity of GLM models.** The GLM model assumes an exponential nonlinearity that governs the mapping from the kernel output (the result of the linear convolution k ⊗ x) to the time-varying spiking rate λt. To assess if this is a valid assumption, we ran the GLM fitting procedure for a given neuron, binned the kernel output across the recorded trials and calculated the observed spike rate for each of these bins, along with the result of applying an exponential non-linearity over the kernel output bins. This analysis showed that the exponential non-linearity is an adequate approximation across the range of kernel output bins observed in the data, with some deviations in extreme and under-sampled bins (see neuron examples in Supplementary Fig. 7a).

**Evaluating the kernel estimation accuracy of GLM models.** GLM encoding models aim to quantify the dependencies of the neural response on multiple coexistent (stimulus, behavioural and task-related) variables. As pointed out before[33], randomness and variability in the timings of these experimental variables are essential for disentangling the different components of the neuronal response. If the relative timings of a given pair of variables was constant across trials, it would be impossible to dissociate the temporal evolution of the firing rate components associated with each of these two variables. For example, in a GO-NOGO task were the animal can collect a reward at a fixed short time after being stimulated with an odour, one cannot disentangle the temporal course of a pre-GO neuronal modulation from the odour response component. In the present virtual reality task, the timing of events is under operant control of the animal, resulting in important timing variability throughout trials. Nevertheless, some degree of temporal correlation between variables could remain, impacting negatively in the ability of the model to discriminate between the contributions of these variables.

To assess the effectiveness of our GLM model in quantifying the contributions of variables to the neuronal response given the recorded data during the task, we used simulated data.

Using an artificial set of known kernels $\boldsymbol{k}^s$ and the set of task variables recorded during real trials $\boldsymbol{x}$, we simulated a spike-count dataset $\boldsymbol{r}_t^s$ that we regressed with the GLM encoding model, and evaluated the accuracy of the model's estimates of the kernels, $\hat{\boldsymbol{k}}^s$. This would allow us to determine the effectiveness of both the kernel inference procedure and the statistical framework for model selection, when the model is faced with the recorded task data $\boldsymbol{x}$ (and taking into account the task design).

To simulate the spiking rate of a neuron $\lambda_t^s$, first we convolved the recorded $\boldsymbol{x}$ (also called 'design matrix') with a random combination of kernels $\boldsymbol{k}^s$ (by randomly choosing a set of kernels from the pool of kernels inferred when analysing real recorded spike-count data), and applied an exponential non linearity to the output of the convolution:

$$\lambda_t^s = \exp\left( \sum_i (k_i^s \otimes x_i)_t \right) \qquad (11)$$

We then simulated $\boldsymbol{r}_t^s$ as a time series of 10-ms-binned spike-count dataset by sampling from a Poisson distribution with parameter $\lambda_t^s$. We simulated 100 neuronal responses from task data $\boldsymbol{x}$ of both first-session and expert-session recordings.

Using $\boldsymbol{r}_t^s$ we performed the cross-validated model selection procedure described before to obtain the $\hat{\boldsymbol{k}}^s$ estimates, which we compared with $\boldsymbol{k}^s$. The correlation between $\boldsymbol{k}^s$ and $\hat{\boldsymbol{k}}^s$ was high across all neuronal simulations (Supplementary Fig. 7b; median correlation of 0.95 and 0.951 for simulations based on first-session and expert-session data), and the correlation between the multiplicative neuronal gain of individual $\boldsymbol{k}^s$ and $\hat{\boldsymbol{k}}^s$ coefficients was also high (Supplementary Fig. 7c). Inspection of individual kernels showed that the simulated kernels were accurately estimated (Supplementary Fig. 7d), and discrepancies in which kernels were not detected by the kernel estimation procedure (i.e., included in $\boldsymbol{k}^s$ simulations but missing in $\hat{\boldsymbol{k}}^s$) or erroneously estimated (i.e., not included in $\boldsymbol{k}^s$ simulation but estimated in $\hat{\boldsymbol{k}}^s$) were typically for kernels of multiplicative gain -1 which have negligible impact on neuronal response encoding (see kernels labelled with asterisks in (Supplementary Fig. 7d). Altogether these results indicate that the model can accurately estimate kernels from the recorded data, capturing with precision the individual contributions of stimulus, behavioural and task-related variables to neuronal responses.

**Contribution of a task variable to the GLM encoding model.** We estimated the contribution of a given variable $x_i$ (associated with a kernel $k_i$) to the encoding model of a neuron by calculating the difference in model performance (spike-normalised log-likelihood increase) between the selected model θ (with firing rate $\lambda_t$) and the model $\theta^{\sim k_i}$ (with firing rate $\lambda_t^{\sim k_i}$) that contains all variables in the selected model, minus the variable $x_i$:

$$Contribution\ of\ variable\ x_i = \frac{\sum_{t \in \{t_{k_i}\}} \left[ (r_t \log(\lambda_t \Delta) - \lambda_t \Delta) - (r_t \log(\lambda_t^{\sim k_i} \Delta) - \lambda_t^{\sim k_i} \Delta) \right]}{\log(2)(1 + \sum_{t \in \{t_{k_i}\}} r_t)} \qquad (12)$$

Here the sum is performed over the set of time bins $\{t_{k_i}\}$ in which the convolution $(k_i \otimes x_i)$ was performed, that is the moments in time were the kernel $k_i$ contributed to the firing rate $\lambda_t$. The normalising factor has an offset of 1 in the sum of $r_t$ over $\{t_{k_i}\}$ to avoid numerical undefined expressions when no spikes were emitted during $\{t_{k_i}\}$, and the log(2) is included to measure the spike-normalised log-likelihood increase in terms of bits. The Contribution of variables calculated for all recorded neurons is shown in Supplementary Fig. 5d. For the percentual Relative contribution of each variable (Fig. 4c), for each neuron we normalised each variable contribution by the sum of all contributions:

$$Relative\ contribution\ of\ variable\ x_i = \frac{Contribution\ of\ variable\ x_i}{\sum_i Contribution\ of\ variable\ x_i} \qquad (13)$$

**Decoding with GLM model.** Decoding with the fitted GLM models, allows trial decoding of binary variable categories, while taking into account neuronal response modulations related to all the other task variables. As was previously shown[7], under this GLM model, the

computation for decoding the identity of a variable $x_j$ with two categories $x_j = A$ and $x_j = B$ (e.g., rewarded or unrewarded odour) with kernels $k_i^A$ and $k_j^{AB}$, respectively, is rather straightforward. Consider the log likelihood ratio (*LLR*) for this situation:

$$LLR = log \frac{likelihood_{x_j=A}}{likelihood_{x_j=B}} = log \frac{p\left(\boldsymbol{r}|\boldsymbol{x}^{\sim j}, x_j=A\right)}{p\left(\boldsymbol{r}|\boldsymbol{x}^{\sim j}, x_j=B\right)} \quad (14)$$

where $\boldsymbol{x}^{\sim j}$ is a vector with the rest of the variables other than $x_j$ (i.e., $\boldsymbol{x} = \{\boldsymbol{x}^{\sim j}, x_j\}^{\sim j}$). Further developing this formulation leads to:

$$LLR = \sum_{t=0}^{T} \left[ r_t \left( \left( k_i^A \otimes x_j \right)_t - \left( k_i^B \otimes x_j \right)_t \right) + \Delta(\lambda_t^A - \lambda_t^B) \right] \quad (15)$$

where $\lambda_t^A$ and $\lambda_t^B$ are the time-varying spiking rate for the model with $x_j = A$ and $x_j = B$ respectively. The *LLR* leads to a simple rule for decoding the category of the variable $x_j$: when $LLR > 0$, the recorded neuron indicates that $x_j = A$ that is more probable under the model; on the contrary, $x_j = B$ is more probable if $LLR < 0$. This mathematical procedure can be conceptualised as a time-varying estimation of whether the emitted spikes up to that time point in the trial are more probable under the spike rate prediction of a model that considers one of the variable categories rather than the alternative. When considering a population of neurons recorded simultaneously, the *LLRs* of the individual neurons can be summed together. In Fig. 5a, this decoding accuracy is averaged across animals. For the analysis of the relation between the accuracy and the number of neurons used for decoding (Supplementary Fig. 10a–c), pseudopopulations of neurons were constructed with subsets of neurons that were randomly sampled from across recording sessions, up to the total number of 117 neurons (matching the total number of neurons from first-session and expert-session recordings). For each population size, the sampling procedure was repeated 50 times and accuracy was averaged. For the analysis shown in Supplementary Fig. 10f aimed at evaluating the contribution to odour decoding accuracy of non-olfactory information encoded in PCx neurons, we calculated *LLRs* where $A$ was set to $O_R$ and $B$ to $O_U$, and $\lambda_t^A$ and $\lambda_t^B$ were calculated in two different ways: one using the full models (i.e., including all fitted kernels) and another after removing contributions of non-odour-related kernels from the models of odour-responsive neurons (i.e., removing non-olfactory modulations from neuron models that had olfactory kernels).

Expert-session recordings typically contained more trials than first-session recordings. To control for the effect of this difference on model estimation and hence accuracy of decoding, we re-fitted GLM models after matching the number of trials used from both training conditions. All decoding results are shown after this trial matching procedure.

In Fig. 5b we compare p($O_R$ | spikes) and p($C_R$ | spikes) with the behavioural response delay (in GO trials the latter corresponds to the reaction time to first lick after first odour inhalation, while in NO GO trials it corresponds to the delay to leave the context zone after first odour inhalation. The quantity p($O_R$ | spikes), the posterior probability of $O_R$ given the observed spikes, is a GLM-based decoding of the PCx representation of $O_R$ in a trial. Following Park et al.[33], we calculated p($O_R$ | spikes) from the GLM encoding distribution using Bayes' rule. If we assume the prior probabilities of $O_R$ and $O_U$ are equal (which is reasonable, since both odours were presented with equal probabilities across trials), then p($O_R$ | spikes) is obtained by:

$$p\left( O_R | r, \left\{ x^{\sim odours}_i \right\} \right) = \frac{p(r|O_R, \{x^{\sim odours}_i\})}{p(r|O_R, \{x^{\sim odours}_i\}) + p(r|O_U, \{x^{\sim odours}_i\})} \quad (16)$$

where on the right side we observe the GLM encoding models and {$x^{\sim odours}_i$} is the set of all covariates except the odour ones. For p($C_R$ |

spikes) a similar formula holds, replacing $O_R$ and ~odours by $C_R$ and ~context.

## Decoding odours with linear classifiers

For the odour decoding analysis of Passive vs. Task conditions (Fig. 6b) and the evaluation of the contribution to odour decoding accuracy of non-olfactory information encoded in PCx neurons (Fig. 6c), we used linear classifiers based on L2-regularised logistic regression with 3-fold cross-validation. This analysis is independent of the GLM models or GLM-based decoding. This guarantees that the obtained decoding accuracies will not be influenced by differences across conditions in the goodness-of-fit or expressive power of GLM models.

Neuronal responses were concatenated into a Trials (T) × Neuron (N) matrix, arranged so that each row contained trial responses to a particular odour. Each neuron's response was averaged over a 0.5-s window after first odour inhalation (for the Passive condition, peak decoding was reached later, at 1 s, so we used that time window). Trials were labelled according to the odour sampled, and arranged into a vector of T elements.

Pseudopopulations of neurons were constructed with subsets of neurons that were randomly sampled from across recording sessions, up to the total number of 117 neurons (matching the total number of neurons from first-session and expert-session recordings). Because different trial conditions contained different numbers of trials, a random subset of responses from each neuron was excluded so that the number of trials per condition was consistent across neurons in the pseudopopulation. For each population size, this trial sampling procedure was repeated 50 times and the obtained accuracies were averaged.

For the analysis of Fig. 6c, a total number of 48 trials were used. For the Passive vs. Task conditions a total number of 25 trials were used (Fig. 6b; the fewer number of trials explains the lower performance reached in Fig. 6b compared to Fig. 6c).

## Clustering groups of neurons with similar relative contributions of their encoded variables

To group neurons according to the task variables that modulate their activities, we used an agglomerative hierarchical clustering approach. We only considered neurons whose GLM encoding model contained at least one task variable (i.e., with a total number of modulating variables ≥1; first session: 115 out of 177 neurons, 65.0%; expert session: 75 out of 117 neurons, 64.1%). First, for each neuron $k$ we determined the vector ***RelContrib***$_k = \{Relative\ contribution\ of\ variable\ x_i\}_i$ containing relative contributions of all variables to the neuron $k$. We then used a correlation metric to calculate the pairwise distances between all vectors of relative contributions, $p\text{dist}_{i,j} = 1 - r_{i,j}$, where $r_{i,j}$ is the sample correlation between the pair (***RelContrib***$_i$, ***RelContrib***$_j$). With these distances we built an agglomerative hierarchical cluster tree, where distances between clusters were specified by the unweighted average distance between all pairs of objects in any two clusters (UPGMA). The obtained hierarchical trees are shown in Supplementary Fig. 9a, d, and the vectors of relative contributions sorted according to their corresponding clusters are shown in Supplementary Fig. 9b, e. To cut a hierarchical tree into individual clusters, we grouped tree leaves using a cutting threshold *thresh$_{cut}$* for the distance coefficients of nodes in the tree (i.e., cutting through the tree at nodes whose heights were lower than *thresh$_{cut}$*; clusters with less than 3 neurons were discarded). The obtained clusters of relative contributions are shown in Fig. 4e and Supplementary Fig. 9c, f with a colour code that is maintained in Supplementary Fig. 9.

We determined the optimal value for *thresh$_{cut}$* by cross-validation of the generalisability of the resulting clusters. In order to do this, we randomly divided trials into 2 sets of equal size (sets A and B) and separately fitted the GLM model to each trial set, obtaining the vectors of the relative contributions of all variables to neuron $k$ under the set $A$ and under the set $B$, ***RelContrib***$^A_k$ and ***RelContrib***$^B_k$ correspondingly.

We first evaluated how well clusters in set *A* generalised to set *B*: we labelled *A* as the train set, and *B* as the test set. We then fixed $thresh_{cut}$ on a given value, and performed the described hierarchical clustering procedure for both $\mathbf{RelContrib}^A_k$ and $\mathbf{RelContrib}^B_k$, obtaining a partitioning matrix for each set, $P^A_{Train}$ and $P^B_{Test}$ (if *N* is the total number of neurons, the partitioning matrix *P* is a binary *N*x*N* matrix, where matrix position $P_{i,j}$ is set to 1 if neuron *i* and neuron *j* belong to the same cluster, and set to 0 otherwise). We then constructed an additional partitioning matrix $P^{A->B}$ by using the centroids of clusters in $P^A_{Train}$ to perform clustering on set *B*: we assigned neuron *k* to cluster *j* in $P^{A->B}$ if $\mathbf{RelContrib}^B_k$ was closest to the centroid of cluster *j* across all cluster centroids in set *A*. To quantify generalisability of clustering results, we calculated the sample correlation coefficient between the upper triangular block of matrix $P^B_{Test}$ and the upper triangular block of matrix $P^{A->B}$. We repeated this procedure using *B* as train set and *A* as test set, and varied $thresh_{cut}$ along a range of values. Through this procedure, we determined that the optimal $thresh_{cut}$ was 2.2 for first-session data, and 2 for expert-session data. For these values of $thresh_{cut}$, generalisability of first-session data was $0.79 \pm 0.006$ (for shuffled test partitionings it was $0.002 \pm 0.032$) and of expert-session data was $0.69 \pm 0.09$ (for shuffled test partitionings it was $0.003 \pm 0.030$).

To visualise the organisation of the relative contribution vectors of all neurons, and perform a qualitative validation of the obtained clusters, we used multidimensional scaling (MDS). This nonlinear technique maps the collection of neuron's relative contribution vectors $\{\mathbf{RelContrib}_i\}_i$ from their original eight-dimensional space (corresponding to the 8 task variables) to a bidimensional projection that best preserves the original distances between vectors $\{pdist_{i,j}\}_{i,j}$. Neurons with similar relative contribution vectors occupy similar regions of the bidimensional MDS plane, allowing visualisation of the global structure of similarities of $\{\mathbf{RelContrib}_i\}_i$. We applied the non-classical (non-metric) MDS algorithm on the collection of $pdist_{i,j}$ values, calculated with the Euclidean distance metric, with Kruskal's normalised stress-1 goodness-of-fit criterion. In Fig. 4e, neurons in the MDS plane were coloured according to the cluster to which they belong, as determined by our hierarchical clustering approach (same colour-code used in Supplementary Fig. 9).

**Reporting summary**

Further information on research design is available in the Nature Portfolio Reporting Summary linked to this article.

## Data availability

Data for this paper are available in https://figshare.com/articles/dataset/piriformData_mat/25944577. The complete data set will be made available upon reasonable request to the corresponding authors. Source data are provided with this paper.

## Code availability

Custom-made Matlab (Mathworks) scripts for GLM analysis are available at https://github.com/marinburginlab/olfactionGLM.

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

## Acknowledgements

We thank members of the Marin-Burgin lab, the Refojo lab and Muraro lab for insightful discussions. We thank Emilio Kropff and Massimo Scanziani for valuable comments on the manuscript. We thank Karel Svoboda and Hidehiko Inagaki for the initial instruction for the in vivo recordings; also to Miho Inagaki for the initial assistance in behavioural training of head-fixed mice; and to the Janelia HHMI Institute for hosting NF. We acknowledge International Development Research Centre IDRC108878 (A.M.B.), Argentine Agency for the Promotion of Science and Technology PICT2018-0880 (A.M.B.), PICT2020-00360 (A.M.B.), PICT 2020-1536 (N.F.), PICT 2016-2758 (N.F.), PICT 2017-4023 (S.A.R.), CONICET PIP 2787 (N.F. and S.A.R.), Swiss National Science Foundation (SNSF) SPIRIT 216044 (A.M.B.) and FOCEM-Mercosur COF 03/11. Fulbright Foundation and 2015-BECar-Argentine Presidential Fellowship in Science and Technology for financial support (N.F.). We thank Ivan Refojo for the help with the 3D printer.

## Author contributions

The project was originally conceptualised by A.M.-B., N.F. and S.A.R. The behavioural paradigm was developed and designed by N.F., S.A.R. and A.M.-B. Task-related hardware was developed and constructed by S.A.R. and N.F. Task-related software was developed and implemented by S.A.R. Animal training, behaviour data collection and behavioural data analysis was performed by N.F., S.A.R., M.A.-D. and L.S. Neural recordings were performed by N.F., S.A.R., L.S. and A.M.-B. Neural data analysis was performed by N.F. and S.A.R. The manuscript was written by N.F., S.A.R. and A.M.-B., and edited and reviewed by all authors.

## Competing interests

The authors declare no competing interests.
