## [Peer Review File · Nature Communications]

Acquisition of non-olfactory encoding improves odour discrimination in olfactory cortexREVIEWER COMMENTS

Reviewer #1 (Remarks to the Author):

In this study, the authors employed a multi-model virtual reality system to investigate how the mouse piriform cortex processes contextual information. They found that mice can quickly learn to connect a specific smell with a visual cue in their environment. The researchers recorded the firing activity of piriform cortex neurons in two groups: one consisted of mice that were new to the virtual reality setup, while the other group comprised mice that had become experts in the odor-task association. Using advanced computational techniques, they revealed that the neurons in the piriform cortex become less selective for the odor stimulus over time, effectively mixing in more task- or context-specific variables.

Despite the piriform cortex's presumed central role in odor learning and memory, there have been limited studies on how piriform population activity changes during learning in awake, behaving mice. In previous studies, that indirectly assessed learning using less advanced analytical methods and simpler go/no-go tests, no substantial changes or representations of reward were observed (e.g., Millman & Murthy, 2020; Wang, Boboila, et al., 2020). Hence, this research is both timely and significant, as it makes a valuable contribution to the field. Additionally, the study is thorough and rigorous, and I have only minor comments and suggestions for improving the manuscript.

1. The text needs to be edited for clarity and grammatical errors.

2. The authors examine activity when the mice enter the context location. However, they can see the grey or green cue in the distance. Thus, context starts when they first see the cue, not when they arrive in the context zone. This is reflected in the separation of Cr and Cu activity in the PCA analyses (Ext. Fig. 2d, components 3, 4 & 9) which separate before context entry. This is worth mentioning in the Discussion.

3. In the text from lines 10 to 15, the authors could provide more detail on how the variables changed during the experiment. This would help readers understand that the changes are adaptive to the task. For example, they might highlight how expert animals slow down and start licking for OrCr, indicating their learning of the odor-context association. They could also speculate on why mice occasionally lick for OrCu on the first day, discussing whether this is due to the dominance of the odor sense or the temporal proximity of the odor cue to the reward.

4. Figs. 1e and 1f: It seems like cells with higher spontaneous firing rates have the biggest change in firing rate (I find “FR variation” confusing). Would the data sort the same way if the authors normalized for mean firing rates and plotted the data using Z scores?

5. Fig. 2a: As the authors state, activity just before odor onset is correlated with the contextual cues in the expert mice but not in the naïve mice, who can presumably also see the cue. Is this interesting or trivial? It would be trivial if this is just due to licking or changes in running speed, etc. However, from Fig. 1, it seems expert mice only start anticipatory licking after odor onset, that expert mice don't lick during OuCr trials, and that the difference in running speed for the Cr vs. Cu mice does not seem dramatically different. Instead, this suggests some sort of heightened arousal and preparatory or anticipatory activity before the potentially rewarded odor arrives. ▸ In the Cu case, the mice may not care about the odor because they already know it will not be rewarded. Can the authors elaborate a bit on this?

6. As I mentioned above, several recent studies (e.g. Wang, Boboila, et al, 2020; Millman & Murthy, 2020) have suggested that the piriform cortex does not encode reward. While at least one of these is likely very wrong (Wang), the authors should at least mention that their data are not consistent with what has been previously reported.

7. In their GLM analyses, the authors find that more variance can be accounted for using their defined parameters in expert vs. first-session mice. The authors would like to attribute this to the mice learning the odor-context association. However, first-session mice need to learn both the task and the odor-context association. Task learning likely contributes substantially to the variability observed in first-session recordings. Ideally, the authors would perform experiments in mice that have learned the task and are then presented with a novel set of contexts and odors. In this case, the only learning required is the association between these two variables. Asking the authors to repeat their experiments using previously trained animals is unfair and unnecessary but, (1) they should outline this in the Discussion, and (2) perhaps there are other variables, such as the variance in trial length or variance in speed along the path, or others, that may reflect task-learning that could be used to explain more of the variance in first-session mice?

8. The authors do not account for the neurons' baseline firing rates. About 5% of piriform neurons are GABAergic interneurons, which have higher firing rates than principal neurons (and are therefore likely oversampled in extracellular recordings). Inhibitory interneurons are more broadly tuned than principal neurons and may be especially context-dependent (e.g. Canto-Bustos et al., 2022). For example, one of the Associative neurons in Fig. 3c has a baseline firing rate of 70 Hz and is strongly suppressed in only the OrCr trials. Can the authors explain away more of their variance if they include baseline firing rate as one of their GLM variables?

9. Fig. 5: Some of the shuffled controls are plotted from 0 to 100% accuracy and others are 50% to 100%. Pick one.

10. The authors compare the sustained ability to decode odor, shown in Fig. 5a, to working memory traces. Do they see anything obviously reminiscent of persistent activity at the level of single neurons? (The example cell shown in Fig. 5b does show some persistent activity, but only in Cr trials. Do they see cells that show persistent activity selectively in CrOr trials?)

Reviewer #2 (Remarks to the Author):

The study by Federman et al. explores how contextual and associative variables are represented in the mammalian olfactory cortex. The authors present a paradigm in which mice are trained to associate odors with visual contexts in virtual reality to obtain a reward while performing electrophysiological recordings in Piriform Cortex (PCx). Following this learning, the authors report that neurons in PCx come to encode non-olfactory features of the task, including visual context, visual-odor pairing, and reward. Single neuron encoding models reveal that odor-responsive neurons acquire mixed selectivity following learning, and contextual variables can be decoded from PCx population activity. Additionally, the authors find that population decoding of odor identity is enhanced following learning. The authors conclude that activity in PCx is modulated by associative learning between odors, contexts, and rewards. This research significantly contributes to our understanding of how sensory responses are modulated by task and behavioral context, and provides a clear demonstration of learned mixed-selectivity in a primary sensory cortex. Given the importance of this work (and as an aside, the paper is beautifully written) I would recommend this manuscript for publication pending the following revisions.

Major Concerns:

1. This study investigates odor-context associations by uniquely associating one specific odor (OR) and context (CR) with a learned lick behavior and reward. The nature of this Go/No-Go paradigm introduces both movement and reward confounds. At the time at which the odor is delivered, the animal knows whether it needs to lick and whether it will be rewarded. Because mice only lick in response to this particular odor-context pairing, it is unclear whether the unique modulation of PCx activity reported on trials of this type constitutes contextual modulation, reward anticipation, motor preparatory activity, or some combination. The authors could more adequately address this concern by controlling for motor and reward anticipatory activity by for example:

- More thorough analysis of PCx activity on false alarm trials, specifically analyzing the mixed-selectivity and population dynamics (as in figure 2)

- Comparing the responses of rewarded odor-context pairs vs rewarded odors (regardless of context)
- Comparing the responses of rewarded odor-context pairs vs rewarded contexts (regardless of odor)
- Presenting the odor before presentation of the reward-predictive cue, allowing comparison of odor responses with and without contextual modulation in trained animals

The contributions of task related variables including licking and reward consumption are addressed in part through the single neuron encoding model but cannot be fully disentangled due to the task design.

2. A major claim of this study is that PCx uses non-olfactory information to ‘enhance odor coding’. However, it is not clear a) that odor coding is enhanced per se (as discussed in the previous comment) and b) that this is due to learned associations between visual contexts and odors. Previous studies (e.g. Chapuis & Wilson 2012, Shakhawat et al. 2014, Calu et al. 2007) have revealed that odor decoding is enhanced following pairing of odors with reward (outside of a learned task context), and that odor identity encoding in PCx becomes more robust and selective over the course of (even passive) odor experience (e.g. Wang & Axel et al. 2020). Without controls addressing these possibilities, it is difficult to evaluate to what extent increases in odor identity decoding are a result of learning the associative odor-virtual context-reward task as the authors claim, as opposed to a consequence of odor experience. The observation that the removal of non-olfactory kernels decreased odor decoding accuracy in odor-encoding neurons may thus point to experimental confounds rather than contributions of non-odor variables to odor coding. The authors address this concern in part by passively presenting odors to two expert animals (Fig.5). However, these responses to passive odors are not compared before and after task learning. Additionally, this manipulation, by removing the animal entirely from the virtual environment eliminates not only contextual information, but also all forms of behavioral engagement. To address these concerns the authors should 1) compare odor responses before and after learning, and 2) switch off the visual context post-learning and conduct an odor-based Go/No-Go task to compare task-induced effects

Minor Concerns

1. The authors report that the acquisition of mixed-selectivity for task-related variables by odor-responsive PCx neurons underlies enhanced odor identity encoding following learning. However, recordings are performed acutely before and after learning in separate animals, precluding tracking of the same cells over the course of learning. It is therefore unclear whether the selectivity of single odor-encoding neurons in PCx is modulated as a result of learning or whether changes in population encoding result from, for example, the emergence of a new population recruited to represent task-relevant information. Greater discussion of the limitations of the study as it pertains to such claims would be appreciated

2. The authors claim that learning enhances odor discrimination but do not directly demonstrate enhanced odor discrimination perceptually or behaviorally (outside of the trained task). Instead,

the authors show that the accuracy of odor identity decoding from population activity is enhanced following learning. Greater precision of language could help to clarify this point

3. The behavioral task described in this study is a spatial task performed in a virtual reality environment. However, the authors do not directly address contributions of spatial information to PCx encoding to motivate this choice. Previous studies (e.g. Poo, et al. 2022l) report that neurons in mammalian PCx display a range of selectivity for odors, spatial locations, and odor-place associations. The authors of the present study note that they find neurons that fire according to the animal's spatial location along the virtual corridor (position responses). To what extent can spatial position be decoded from PCx activity before and after learning? Do neurons encoding positional information constitute a distinct subset of neurons? Are these neurons more likely to show associative responses?

4. Authors should include more explicit discussion of differences in metrics of behavioral engagement (i.e. sniffing rate, running speed) between first session and expert session recordings

5. Changes in PCx dynamics could better be described as emerging “following learning” instead of “with learning” given that activity is only recorded either before or after learning has taken place, and not over the course of task learning in the same animal

6. Previous studies have investigated the question of how various forms of odor-reward learning affect odor representations in PCx and have described enhanced odor identity decoding, as well as encoding of associative and value information in PCx (including among others, Chapuis & Wilson 2012, Shakhawat et al. 2014, Wang.. Axel et al. 2020, Ottenheimer et al. 2022, Calu et al. 2007, Gire et al. 2013, Roesch et al. 2007). The authors should contextualize the findings of the current study within these observations to better highlight the unique nature of the contextual and associative responses described in this study vs. other previously described forms of odor response modulation by reward learning

7. The procedure for generating PSTHs, with alignment to means without any per-trial time warping is idiosyncratic, and not adequately explained in the text. The authors should justify the decision to omit the standard time-warping procedure, and show that the inter-intratrial-event intervals are narrow enough that aligning to median intra-trial events is adequate.

Figure Concerns

● In general, the small colorbars to the sides of the raster plots (e.g. Fig 1e, the green colorbars) are too narrow to be easily visible. Please widen them.

● In Fig 1E, the authors make the unconventional choice to show firing rate variation instead of firing rate. To reduce potential confusion, the authors should consider showing firing rate instead of “FRv” or rename the axes to more accurately represent the method, such as “baseline-subtracted firing rate”

● In Fig 3B, it's unclear whether the subplots are the same neuron or different neurons. The authors should clarify this point in the legend. It could also be useful to show a neuron with highly mixed selectivity (i.e. a neuron with 3 or 4 modulating variables as described in Fig 4d)

Response to reviewers

Note that all changes in the new version of the manuscript are highlighted in bold.

Reviewer #1 (Remarks to the Author):

In this study, the authors employed a multi-model virtual reality system to investigate how the mouse piriform cortex processes contextual information. They found that mice can quickly learn to connect a specific smell with a visual cue in their environment. The researchers recorded the firing activity of piriform cortex neurons in two groups: one consisted of mice that were new to the virtual reality setup, while the other group comprised mice that had become experts in the odor-task association. Using advanced computational techniques, they revealed that the neurons in the piriform cortex become less selective for the odor stimulus over time, effectively mixing in more task- or context-specific variables.

Despite the piriform cortex's presumed central role in odor learning and memory, there have been limited studies on how piriform population activity changes during learning in awake, behaving mice. In previous studies, that indirectly assessed learning using less advanced analytical methods and simpler go/no-go tests, no substantial changes or representations of reward were observed (e.g., Millman & Murthy, 2020; Wang, Boboila, et al., 2020). Hence, this research is both timely and significant, as it makes a valuable contribution to the field. Additionally, the study is thorough and rigorous, and I have only minor comments and suggestions for improving the manuscript.

1. The text needs to be edited for clarity and grammatical errors.

Thanks for the suggestion, we have now edited the text.

2. The authors examine activity when the mice enter the context location. However, they can see the grey or green cue in the distance. Thus, context starts when they first see the cue, not when they arrive in the context zone. This is reflected in the separation of Cr and Cu activity in the PCA analyses (Ext. Fig. 2d, components 3, 4 & 9) which separate before context entry. This is worth mentioning in the Discussion.

We thank the reviewer for the suggestion, indeed as the reviewer highlights, animals can see the approaching visual context from a distance. We mentioned that in the description of the task. We have further emphasized now in the results section Page 6. It is worth mentioning that we performed decoding of contextual information along the entire corridor in Fig. 5a and accurate decoding of visual context information occurs in the proximity of the context zone.

3. In the text from lines 10 to 15, the authors could provide more detail on how the variables changed during the experiment. This would help readers understand that the changes are adaptive to the task. For example, they might highlight how expert animals slow down and start licking for OrCr, indicating their learning of the odor-context association. They could also speculate on why mice occasionally lick for OrCu on the first day, discussing whether this is due to the dominance of the odor sense or the temporal proximity of the odor cue to the reward.

We agree with the reviewer that the section required a better description. We have now expanded the description of the changes in behavior following learning on Page 4.

4. Figs. 1e and 1f: It seems like cells with higher spontaneous firing rates have the biggest change in firing rate (I find “FR variation” confusing). Would the data sort the same way if the authors normalized for mean firing rates and plotted the data using Z scores?

We agree with the reviewer that FR variation is confusing. We modified the labels on the axes to “Mean subtracted FR”. We did not use Z-scores because it tends to amplify the noisy signal from neurons with low variance in firing rate. We use the mean-subtracted firing rate solely for visualization and it is closer to representing raw data.

5. Fig. 2a: As the authors state, activity just before odor onset is correlated with the contextual cues in the expert mice but not in the naïve mice, who can presumably also see the cue. Is this interesting or trivial? It would be trivial if this is just due to licking or changes in running speed, etc. However, from Fig. 1, it seems expert mice only start anticipatory licking after odor onset, that expert mice don’t lick during OuCr trials, and that the difference in running speed for the Cr vs. Cu mice does not seem dramatically different. Instead, this suggests some sort of heightened arousal and preparatory or anticipatory activity before the potentially rewarded odor arrives. In the Cu case, the mice may not care about the odor because they already know it will not be rewarded. Can the authors elaborate a bit on this?

This is indeed interesting and we have now highlighted it in the results section Page 6 and in the discussion.

6. As I mentioned above, several recent studies (e.g. Wang, Boboila, et al, 2020; Millman & Murthy, 2020) have suggested that the piriform cortex does not encode reward. While at least one of these is likely very wrong (Wang), the authors should at least mention that their data are not consistent with what has been previously reported.

We apologize for the missing references. We have now mentioned these studies in the Discussion section.

We found PCx neurons responding to rewards, and to rewarded and non-rewarded associations of odors. This appears to be at odds with previous studies where no substantial changes or representations of reward were observed in PCx (Millman & Murthy, 2020; Wang, Boboila, et al., 2020). However, the behavioural paradigm that we use is a more complex task, where expert animals differentiate the same odour in different visual contexts. Furthermore, the virtual environment allows the animals to operate for the presentation of stimuli and/or rewards, allowing a different level of commitment during the performance of the task than in simpler go/no go tasks, as used in the previous works. This could have enabled us to observe strong neural responses involved, necessary for PCx to encode identity, reward and association. This is now discussed in the Discussion Section Page 16.

7. In their GLM analyses, the authors find that more variance can be accounted for using their defined parameters in expert vs. first-session mice. The authors would like to attribute this to the mice learning the odor-context association. However, first-session mice need to learn both the task and the odor-context association. Task learning likely contributes substantially to the variability observed in first-session recordings. Ideally, the authors would perform experiments in mice that have learned the task and are then presented with a novel set of contexts and odors. In this case, the only learning required is the association between these two variables. Asking the authors to repeat their experiments using previously trained animals is unfair and unnecessary but, (1) they should outline this in the Discussion, and (2) perhaps there are other variables, such as the variance

in trial length or variance in speed along the path, or others, that may reflect task-learning that could be used to explain more of the variance in first-session mice?

We thank the reviewer for the comment. We would definitely explore the suggested experiment in future work. We added a sentence about this to the result section: “The lower reliability of responses observed in first session animals could also be related to a less stereotyped behaviour and underlying neuronal processes due to mice being exposed for the first time to the task” on Page 9.

8. The authors do not account for the neurons’ baseline firing rates. About 5% of piriform neurons are GABAergic interneurons, which have higher firing rates than principal neurons (and are therefore likely oversampled in extracellular recordings). Inhibitory interneurons are more broadly tuned than principal neurons and may be especially context-dependent (e.g. Canto-Bustos et al., 2022). For example, one of the Associative neurons in Fig. 3c has a baseline firing rate of 70 Hz and is strongly suppressed in only the OrCr trials. Can the authors explain away more of their variance if they include baseline firing rate as one of their GLM variables?

We apologize that we may not have been clear enough regarding this point. We indeed included the baseline firing rate as one of the GLM variables. This is stated in the Methods section under the Bias Kernel. We have now also included this in the results: “We also included a bias kernel to account for differences in the baseline firing rate of neurons”.

Regarding the example neuron in Fig 3c, as the reviewer mentioned, it is a very interesting example. If the neuron is indeed an inhibitory cell (which we unfortunately could not confirm with 100% certainty with extracellular recordings), the fact that the neuron hyperpolarize only in rewarded trials suggests a specific release from inhibition (or disinhibition) in that condition, which could potentially promote plasticity. We are very excited about that idea and are planning to further explore such a mechanism in future experiments, perhaps by phototagging inhibitory neurons.

9. Fig. 5: Some of the shuffled controls are plotted from 0 to 100% accuracy and others are 50% to 100%. Pick one.

We have now corrected that.

10. The authors compare the sustained ability to decode odor, shown in Fig. 5a, to working memory traces. Do they see anything obviously reminiscent of persistent activity at the level of single neurons? (The example cell shown in Fig. 5b does show some persistent activity, but only in Cu trials. Do they see cells that show persistent activity selectively in CrOr trials?)

We do see neurons that have persistent activity in CrOr trials; one example is in Extended data Fig. 3i, neuron #3 N59. We included a sentence to highlight that example neuron in the text: “Interestingly, some of the associative neurons that we found show persistent excitatory or inhibitory activity in ORCr trials (Associative neurons in Fig. 3c and Extended data Fig. 3i, see neuron #3 N59).”

Reviewer #2 (Remarks to the Author):

The study by Federman et al. explores how contextual and associative variables are represented in the mammalian olfactory cortex. The authors present a paradigm in which mice are trained to associate odors with visual contexts in virtual reality to obtain a reward while performing electrophysiological recordings in Piriform Cortex (PCx). Following this learning, the authors report that neurons in PCx come to encode non-olfactory features of the task, including visual context, visual-odor pairing, and reward. Single neuron encoding models reveal that odor-responsive neurons acquire mixed selectivity following learning, and contextual variables can be decoded from PCx population activity. Additionally, the authors find that population decoding of odor identity is enhanced following learning. The authors conclude that activity in PCx is modulated by associative learning between odors, contexts, and rewards. This research significantly contributes to our understanding of how sensory responses are modulated by task and behavioral context, and provides a clear demonstration of learned mixed-selectivity in a primary sensory cortex. Given the importance of this work (and as an aside, the paper is beautifully written) I would recommend this manuscript for publication pending the following revisions.

Major Concerns:

1. This study investigates odor-context associations by uniquely associating one specific odor (OR) and context (CR) with a learned lick behavior and reward. The nature of this Go/No-Go paradigm introduces both movement and reward confounds. At the time at which the odor is delivered, the animal knows whether it needs to lick and whether it will be rewarded. Because mice only lick in response to this particular odor-context pairing, it is unclear whether the unique modulation of PCx activity reported on trials of this type constitutes contextual modulation, reward anticipation, motor preparatory activity, or some combination.

We understand the reviewer's concerns and agree with the idea that the activity previous to the decision to lick for reward likely involves a combination of signals. We have analyzed the data in the manuscript in various ways to try to separate each of the variables that could modulate responses of different types of trials.

We believe that at the particular moment in which animals receive the odor in rewarded trials, these three phenomena (contextual modulation, reward anticipation and motor preparation) could occur simultaneously. We have separated those using GLM (see further comments below), but also using other analyses:

Regarding *contextual modulation*, which we interpret as a modulation that occurs exclusively due to the presence of Cr or Cu, we have identified those using dPCA. An example of that can be observed in Extended data Fig. 2D component #9, which separates rewarded context (dark lines) from unrewarded context (light lines).

Regarding *Reward anticipation*, we interpret that as the activity modulation that might be observed in GO trials once the animal smells the odour but which takes place before trial outcome (e.g., before collecting a reward in hit trials or failing to collect it in False Alarm trials). This represents a cognitive-associative signal. We have examples of single neurons that show differential activity in rewarded trials, preceding reward consumption. Such an example can be seen in Fig. 3c in associative neurons and in Extended data Fig. 3i. In addition, those signals can also be seen pre-reward in dPCA that distinguish O_RCR trials from the rest, as shown in Extended data Fig. 2D components #3, #4, #12.

Regarding *motor preparation*, that we interpret as the exclusively motor activity previous to a lick, we agree that those are difficult to separate because in rewarded trials they only appear before receiving the reward. For that, and based on the reviewer's concern, we have further analyzed data by evaluating activity preceding each lick in expert animals, but differentiating licks around the reward delivery zone from licks that occur outside (i.e., before odor stimulation). Neuronal activity preceding the former licks could contain motor preparation and/or reward anticipation signals. On the other hand, activity preceding the latter licks might contain motor preparation signals, but should not be associated with reward anticipation since the animal has not been stimulated with the odor yet. Interestingly, we found different examples of neurons: some neurons show responses in both types of licks but with higher responses in licks that occur in the rewarded zone (showing a mixture of both components in the response at the rewarded zone), while others specifically responded with reward anticipation and did not respond with anticipation to licks that occur outside the rewarded zone (showing a clear difference in reward anticipation vs motor preparation). We have now included a figure with these example neurons as an Extended data Fig 4.

The authors could more adequately address this concern by controlling for motor and reward anticipatory activity by for example:

- More thorough analysis of PCx activity on false alarm trials, specifically analyzing the mixed-selectivity and population dynamics (as in figure 2)

We thank the reviewer for the suggestion. We have now compared false alarm and hit trials and added a panel in Extended data Fig.2e with the results. We analyzed population dynamics with dPCA and also examined PSTH responses. Interestingly, we found similar trajectories in dPCA for false alarm and hit trials before receiving reward, which then diverge after reward delivery. This indicates a similar neuronal process prior to the decision to lick in these type of trials.

As explained above, we have in addition analyzed PSTH of individual neurons that have only reward anticipatory responses, as well as neurons that show only pre-motor responses or both (Extended data Fig. 4).

These support the idea that although the population activity previous to the lick decision has multiple components, single neuron activity can separate individual contributions, as also reflected in our GLM analysis.

- Comparing the responses of rewarded odor-context pairs vs rewarded odors (regardless of context)
- Comparing the responses of rewarded odor-context pairs vs rewarded contexts (regardless of odor)

In the manuscript, we have compared OrCr vs OrCu, which represents the same odour in different contexts, to evaluate contextual modulation using dPCA in Fig 2. We have now added the comparison proposed by the reviewer of rewarded odor-context pairs vs rewarded odors (regardless of context) and conducted odour decoding (Extended data Fig. 10d). We found that decoding accuracy was faster and higher in Cr trials of expert animals compared to all trials regardless of contexts. This further supports the idea that learning induces contextual modulation of odour responses, which increases decoding capabilities. Interestingly, comparing curves of odour decoding across all trials (regardless of context) between first session and expert animals reveals no difference, suggesting that repeated exposure to odours across training does not significantly improve odour decoding in PCx in this case, presumably because these are very

different pure odours that can be accurately decoded in the first session (decoding does not increase due to perceptual learning in this case).

Indeed, we performed additional experiments in which we recorded 93 neurons from 4 first session animals that were stimulated passively with odours. Accuracy of odour identity decoding under passive odour exposure was not different between these first session animals and the expert animals we recorded previously (see Fig 6b, solid and dashed black lines for expert and first session animals, respectively), indicating that repetitive odour exposure throughout training does not impact in PCx odour decoding for the odours used in the task.

- Presenting the odor before presentation of the reward-predictive cue, allowing comparison of odor responses with and without contextual modulation in trained animals

The reviewer suggests performing new experiments to account for the difference in contextual modulation of odors. This is an interesting suggestion that we would like to explore in future experiments. We believe that these experiments will take time and may not significantly change the conclusions of this work. Instead, we explored the contribution of contextual information by comparing passive odor exposure vs exposure during the task.

The contributions of task related variables including licking and reward consumption are addressed in part through the single neuron encoding model but cannot be fully disentangled due to the task design.

This is a valid concern. We used GLM to disentangle the contribution of individual variables to the modulation of neuronal activity at the level of single neurons. Due to the reviewer's concern, we have further evaluated the capacity of our GLM model to disentangle the contribution of variables by assessing the accuracy by which the model can capture the individual kernels when working on simulated data based on behavioral recordings. We have added an Extended data Fig. 7 with this analysis and a text in the methods section (see *Validating the exponential nonlinearity of GLM models* and *Evaluating the kernel estimation accuracy of GLM models* sections in methods). This result proves that the GLM model captures with great accuracy the individual contributions even with correlated variables.

2. A major claim of this study is that PCx uses non-olfactory information to 'enhance odor coding'. However, it is not clear a) that odor coding is enhanced per se (as discussed in the previous comment) and b) that this is due to learned associations between visual contexts and odors. Previous studies (e.g. Chapuis & Wilson 2012, Shakhawat et al. 2014, Calu et al. 2007) have revealed that odor decoding is enhanced following pairing of odors with reward (outside of a learned task context), and that odor identity encoding in PCx becomes more robust and selective over the course of (even passive) odor experience (e.g. Wang & Axel et al. 2020) . Without controls addressing these possibilities, it is difficult to evaluate to what extent increases in odor identity decoding are a result of learning the associative odor-virtual context-reward task as the authors claim, as opposed to a consequence of odor experience. The authors address this concern in part by passively presenting odors to two expert animals (Fig.5). However, these responses to passive odors are not compared before and after task learning. Additionally, this manipulation, by removing the animal entirely from the virtual environment eliminates not only contextual information, but also all forms of behavioral engagement.

To address these concerns the authors should 1) compare odor responses before and after learning, and 2) switch off the visual context post-learning and conduct an odor-based Go/No-Go task to compare task-induced effects

The reviewer raises two main concerns that we address here:

Concern 1: The improvement in decoding accuracy could be a result of repeated exposure to odours throughout training, and not due to an odour-context association.

The improvement in odour decoding accuracy in expert animals is observed when comparing decoding odour identity in Cr vs. in Cu trials. We found that decoding in Cr trials is more accurate than Cu trials (see Extended Fig 10a, and bars “Data” in Extended Fig 10f). This cannot be explained by an effect of repeated exposure to odours through training, since in the comparison performed (Cr vs Cu trials in expert animals) we are comparing the same set of neurons from the same animals (thus, there is no difference in previous odor exposure between Cr and Cu conditions). This effect is the result of a process that varies from trial to trial, related to the odor-context association. Indeed, in unrewarded contexts decoding accuracy of expert animals recapitulates the accuracy observed in passive exposure (Fig 5c), so it is clearly an “active” process that when “engaged” in rewarded contexts improves odour decoding. This dynamic effect cannot be explained by the repeated exposure through training.

Furthermore, as mentioned above, the new experiments of passive odour exposure of first session animals show that, for the odours used in our task, repetitive odour exposure throughout training does not impact in PCx odour decoding (see Fig 6b, solid and dashed black lines for expert and first session animals, respectively).

Last, we added a new analysis proposed by the reviewer comparing rewarded odour-context pairs vs rewarded odours (regardless of context) and conducted dPCA-based odour decoding (Extended data Fig. 10d). Comparing curves of odour decoding across all trials between first session and expert animals reveals no difference, suggesting that repeated exposure to odours across training does not significantly improve odour decoding in PCx in this case. As mentioned above, this could be due to the use of very different pure odours that produce very different activity in PCx already in first session.

Concern 2: We do not test the effect on odour decoding before and after learning, staying in an “engaged” paradigm in which there is no context (i.e., an odor-based Go/No-Go task).

We understand the reviewer concern regarding that we could be missing a potential effect of engagement if we compared decoding accuracies with a passive odour exposure condition. Nevertheless, we performed different analyses to isolate the olfactory neuronal responses from the rest of the neuronal modulations that take place during the virtual reality task.

First, we used dPCA, that disentangles data dependencies on different variables, and separate the contribution of odours from those of context and odour-context interactions during the task. In this way, we can evaluate discriminability of odour responses during the task irrespective of context type. The dPCA algorithm is a well-tested and highly successful method specially designed for this goal. The dPCA odour component was used to decode odour identity throughout time across trials (horizontal black bars indicate periods of significant odour-decoding accuracies in Extended data Fig 2 b,c). We added a graph to show these accuracies peaked at 87% at 0.51s after first odour inhalation and 88.5% at 0.53s after first odour inhalation for first-session and expert animals, respectively (see “All trials” curves in Extended data Fig. 10d). This shows that, removing contextual contributions, odour decoding does not improve significantly after repeated exposure. There is, however, a significant decoding improvement specifically in the subset of Cr trials of

expert animals (98.5% at 0.36s after first odour inhalation, see “Cr trials” curves in Extended data Fig. 10d).

Furthermore, engagement alone does not seem enough to improve odour decoding accuracy at the levels observed in Cr trials of expert animals. We used linear decoders on the recorded neuronal responses and observed that while in Cr trials accuracies reach 95.3%, comparison of odour decoding in expert animals between Cu trials (engaged) and passive stimulation trials (disengaged) results in 82% and 77.5% accuracies, respectively (see Fig 5d). Again, while there could be a minor effect of engagement, overall this indicates that the association with the rewarded context is responsible for the improvement, rather than engagement alone (Note: subtle differences in the accuracy values obtained with dPCA, shown in Fig 10d, and in the analysis shown in Fig 5d is due to the fact that the set of neurons analysed is not the same, since different animals were used in both analyses).

Finally, in Extended data Fig. 10f, we used linear decoders on GLM-based neuronal activity simulations, which allows us to discriminate between the contributions of the different variables to the odour decoding accuracy. In the “Only odour” bars of Extended data Fig. 10f we show the decoding accuracy obtained when isolating the olfactory neuronal responses from the rest of the neuronal modulations (using only the contribution of odor kernels), while the animal is engaged in the task. The odour decoding accuracies were no different between first-session and expert animals, in both Cu and Cr trials (Extended data Fig. 10F “Only Odour” bars), indicating that repeated odour exposure has not changed discriminability of pure odour responses (this agrees with the results cited above for decoding of passive odours in first session and expert animals, Fig 6b). Nor can these olfactory responses explain the improvement in decoding in rewarded contexts when the animal is expert. So improvement of decoding accuracies of expert animals in Cr trials is not explained by purely olfactory responses.

Overall, all analysis point to the same conclusions: in our experiments odour responses do not become more discriminable by repeated odour exposure alone, engagement has a minor effect on this discriminability, and in Cr trials expert animals show an enhancement in odour decoding accuracy that is related to the learned association.

Finally, we have added a paragraph in the discussion regarding previous studies on the effect of learning in odor processing, as suggested by the reviewer.

Additional comment: the number of expert animals used for the analysis of odour decoding in task vs passive exposure was 4 and not 2 (Fig 6b). We made a mistake in the description of the experiment in *Methods*, where we said we used 2 animals, we apology for that. It has been corrected now.

The observation that the removal of non-olfactory kernels decreased odor decoding accuracy in odor-encoding neurons may thus point to experimental confounds rather than contributions of non-odor variables to odor coding.

We would like to clarify our point of view regarding this particular remark by the reviewer, since it is central to our claims. We think that the term “experimental confound” to refer to variables other than odor might lead to a misinterpretation of the results. To avoid confusions and make our point clear, we would like to explicitly define what we mean. We will base our discussion according to the review “How to control for confounds in decoding analyses of neuroimaging data” by Snoek and collaborators (REF)

When performing a statistical analysis of the relationship between a dependent variable Y and an independent variable X, one should control for potential confounding variables C that could lead

to spurious statistical relationships between X and Y. These spurious results could arise if C is correlated with both X and Y: in this scenario, any statistical relation between X and Y may be completely explained by variable C, and not by a real relation between X and Y (that is, when C is taken into account the correlation between X and Y vanishes).

In the case of the analysis of odor-identity decoding in PCx (Fig 5d), using the category “confounding variable” to refer to PCx activity resulting from non-olfactory variables is misleading. These non-olfactory variables would indeed be confounding if these variables were scientifically uninteresting/uninformative to the hypothesis being tested. The tested hypothesis is: “modulations in PCx activity unrelated to odors affect the decodability of odor identity in the PCx”. Thus, the non-olfactory variables are not “confounding”, but are additional covariates that act together with X (odor-related modulations) to inform predictions of Y (odor identity decoding).

As discussed by Snoek and collaborators (How to control for confounds in decoding analyses of neuroimaging data. Snoek et al, NeuroImage 2019), to disentangle the contribution of each underlying variable to the accuracy of a decoding analysis can be challenging. As the authors show (see “1.2.3 Control for confounds during pattern estimation”), one effective approach is to exploit an encoding GLM model to regress the contributions of the variables to neuronal activity and then perform decoding on GLM simulations where the contributions of variables are turned off by disregarding their corresponding kernels when performing the activity simulations. This is the approach we implemented in Fig. 6c and Sup Fig 10f.

Results of these analysis further support that the non-odorant kernels do not constitute confounding variables, but are variables that contribute additional information for the odor decoding procedure. If they were confounding variables responsible for a spurious relationship between odor kernels and odor decoding, when removed from GLM simulations odor identity decoding should approach chance levels. This is not at all what we observe, and accuracy approaches 90% (See “only odor” bars in Sup Fig 10f). This is absolutely reasonable to expect, since we are using PCx odor kernels to decode odor identity. The interesting point of this analysis is that the non-odorant kernels contribute additional information, improving odor identity decoding specifically in rewarded contexts.

We consider these results particularly important, since encoding of non-sensory information in primary sensory cortices could in principle interfere with the sensory function of these brain regions. We find that in the PCx it is quite the opposite, enhancing its olfactory function in a context-dependent manner.

Ref: How to control for confounds in decoding analyses of neuroimaging data. Snoek et al, NeuroImage 2019.

Minor Concerns

1. The authors report that the acquisition of mixed-selectivity for task-related variables by odor-responsive PCx neurons underlies enhanced odor identity encoding following learning. However, recordings are performed acutely before and after learning in separate animals, precluding tracking of the same cells over the course of learning. It is therefore unclear whether the selectivity of single odor-encoding neurons in PCx is modulated as a result of learning or whether changes in population encoding result from, for example, the emergence of a new population recruited to represent task-relevant information. Greater discussion of the limitations of the study as it pertains to such claims would be appreciated.

We thank the reviewer for the suggestion. We added a sentence related to this concern in the results section “Learned mixed-selectivity responses on PCx are structured”.

2. The authors claim that learning enhances odor discrimination but do not directly demonstrate enhanced odor discrimination perceptually or behaviorally (outside of the trained task). Instead, the authors show that the accuracy of odor identity decoding from population activity is enhanced following learning. Greater precision of language could help to clarify this point.

We replaced discrimination by decoding in several places of the manuscript to avoid confusion. We have also added a new analysis with a corresponding Figure to evaluate if a better single-trial odour and context decoding is reflected in faster behavioural trial discrimination (Fig. 5b,c). For that we analyzed the temporal delay to perform a behavioural response and correlated it with decoding accuracy of rewarded context and rewarded odour. The results show a significant correlation, indicating that the association of odour and context encoding in PCx predicts behavioral discrimination performance.

3. The behavioral task described in this study is a spatial task performed in a virtual reality environment. However, the authors do not directly address contributions of spatial information to PCx encoding to motivate this choice. Previous studies (e.g. Poo, et al. 2022l) report that neurons in mammalian PCx display a range of selectivity for odors, spatial locations, and odor-place associations. The authors of the present study note that they find neurons that fire according to the animal's spatial location along the virtual corridor (position responses). To what extent can spatial position be decoded from PCx activity before and after learning? Do neurons encoding positional information constitute a distinct subset of neurons? Are these neurons more likely to show associative responses?

Our analysis was similar to that of Poo et al., in which we decoded context identity in different positions, while their study decoded odour port identity from 4 cardinal locations. In our study, decoding context had different accuracy depending on the position (Fig. 5a), increasing as the animal arrives to the visual context. Spatial position, as a continuous variable, was not decoded in our study, nor was it in Poo et al.

Regarding the neurons that carry contextual information, we observed that they tend to group with responses to odors (Fig. 4e), indicating an association of odours and contexts. We never observed neurons that show modulation only to contexts.

4. Authors should include more explicit discussion of differences in metrics of behavioral engagement (i.e. sniffing rate, running speed) between first session and expert session recordings.

We agree with the reviewer that the section required a better description. We have now expanded the description of the changes in behavior with learning in Page 4.

5. Changes in PCx dynamics could better be described as emerging "following learning" instead of "with learning" given that activity is only recorded either before or after learning has taken place, and not over the course of task learning in the same animal.

We have now changed that in the text.

6. Previous studies have investigated the question of how various forms of odor-reward learning affect odor representations in PCx and have described enhanced odor identity decoding, as well as encoding of associative and value information in PCx (including among others, Chapuis & Wilson 2012, Shakhawat et al. 2014, Wang.. Axel et al. 2020, Ottenheimer et al. 2022, Calu et al. 2007, Gire

et al. 2013, Roesch et al. 2007). The authors should contextualize the findings of the current study within these observations to better highlight the unique nature of the contextual and associative responses described in this study vs. other previously described forms of odor response modulation by reward learning.

We apologize for the missing references. We have now mentioned the studies in the discussion.

7. The procedure for generating PSTHs, with alignment to means without any per-trial time warping is idiosyncratic, and not adequately explained in the text. The authors should justify the decision to omit the standard time-warping procedure, and show that the inter-intratrial-event intervals are narrow enough that aligning to median intra-trial events is adequate.

We have also tried using time-warping to align PSTHs and found similar results to time-stitching, overall. We chose time-stitching, as others have done before (Park IM et al, Nature Neuroscience 2014), because time-warping procedures tended to distort the firing rate time course in some trials in which the inter-intratrial-event intervals were long.

Figure Concerns

- In general, the small colorbars to the sides of the raster plots (e.g. Fig 1e, the green colorbars) are too narrow to be easily visible. Please widen them.

Done

- In Fig 1E, the authors make the unconventional choice to show firing rate variation instead of firing rate. To reduce potential confusion, the authors should consider showing firing rate instead of “FRv” or rename the axes to more accurately represent the method, such as “baseline-subtracted firing rate”

We have renamed the axes as mean subtracted FR.

- In Fig 3B, it’s unclear whether the subplots are the same neuron or different neurons. The authors should clarify this point in the legend. It could also be useful to show a neuron with highly mixed selectivity (i.e. a neuron with 3 or 4 modulating variables as described in Fig 4d)

They are different neurons; we have now clarified that in the legend.

REVIEWERS' COMMENTS

Reviewer #1 (Remarks to the Author):

The authors have put together a very elegant study that has been improved substantially by incorporating the suggestions that Reviewer 2 and I had from the initial submission. I think the authors have gone above and beyond to address all my concerns, and I trust that Reviewer 2 feels the same way. The manuscript is of great quality and publishing this paper will be a tremendous coup for Nature Communications.

Reviewer #2 (Remarks to the Author):

This is an excellent revision of a paper I was previously enthusiastic about. The extensive additional analysis to better define the influence of passive experience and to disentangle task variables makes the paper more convincing, and it is significantly cleaned up relative to the initial submission. I congratulate the authors on this nice work, and think that the paper is now appropriate for publication.